# Ecological analysis of Pavlovian fear conditioning in rats

Peter R. Zambetti [1], Bryan P. Schuessler[1], Bryce E. Lecamp[2], Andrew Shin [3], Eun Joo Kim [1] & Jeansok J. Kim [1✉]

Pavlovian fear conditioning, which offers the advantage of simplicity in both the control of conditional and unconditional stimuli (CS, US) presentation and the analysis of specific conditional and unconditional responses (CR, UR) in a controlled laboratory setting, has been the standard model in basic and translational fear research. Despite 100 years of experiments, the utility of fear conditioning has not been trans-situationally validated in real-life contexts. We thus investigated whether fear conditioning readily occurs and guides the animal's future behavior in an ecologically-relevant environment. To do so, Long-Evans rats foraging for food in an open arena were presented with a tone CS paired with electric shock US to their dorsal neck/body that instinctively elicited escape UR to the safe nest. On subsequent test days, the tone-shock paired animals failed to exhibit fear CR to the CS. In contrast, animals that encountered a realistic agent of danger (a looming artificial owl) paired with a shock, simulating a plausible predatory strike, instantly fled to the nest when presented with a tone for the first time. These results highlight the possibility of a nonassociative, rather than standard associative, fear process providing survival function in life-threatening situations that animals are likely to encounter in nature.

[1] Department of Psychology, University of Washington, Seattle, WA 98195, USA. [2] Undergraduate Program in Neuroscience, University of Washington, Seattle, WA 98195, USA. [3] Undergraduate Program in Human Biology, Stanford University, Stanford, CA 94305, USA. ✉email: jeansokk@u.washington.edu

Since the time of Watson and Morgan's[1] conception that emotions, such as fear, should be studied as conditional (acquired) reactions and Watson and Rayner's[2] demonstration that fear can be rapidly learned in 9-month-old "Little Albert," Pavlovian (or classical) fear conditioning has been the paradigm par excellence for studying both normal and abnormal fear behaviors[3–7]. Briefly, fear conditioning focuses on how an initially innocuous conditional stimulus (CS; e.g., auditory, visual, contextual cues), upon pairing with a noxious unconditional stimulus (US; usually electric shock) that reflexively elicits unconditional responses (UR; namely defensive reactions), becomes capable of eliciting conditional responses (CR; e.g., freezing in rodents, increased skin conductance in humans). A century of fear conditioning research has led to wide-ranging discoveries. In particular, fear conditioning experiments have fundamentally transformed learning theories from the archaic contiguity (or temporal) relationship[8–10] to the modern contingency (or informational) relationship between the CS and US[11–14], revealed detailed neurobiological mechanisms of learning and memory[15–17] and influenced contemporary cognitive behavioral therapy for various anxiety and trauma-related disorders, such as panic, phobic and posttraumatic stress disorders[18–22].

Despite the utility and appeal of fear conditioning paradigms, in particular the fact that conditional fear memory can transpire after *a single* CS-US pairing and be retained across the adult lifespan[23,24], they nonetheless simplify behavioral analyses of fear, ignoring the multitude of actions and decisions that animals and humans utilize to survive the breadth of risky situations in the real world[25–30]. Indeed, standard rodent fear conditioning studies performed in small experimental chambers encapsulate Thorndike's notion of studying unadulterated learning by placing animals in artificial situations that inhibit "instinctive activities (e.g., instinctive fears)," as instinctive behaviors may be opposite to learned behaviors in complex environments[31]. Hence, the prevalent notion that fear conditioning produces biologically functional associative fear memory needs to be ecologically validated. In fact, some researchers have questioned the evolutionary logic underlying fear conditioning; "No owl hoots or whistles 5 seconds before pouncing on a mouse…Nor will the owl give the mouse enough trials for the necessary learning to occur…What keeps animals alive in the wild is that they have very effective innate defensive reactions which occur when they encounter any kind of new or sudden stimulus"[32]. Consistent with this contrarian view are findings that laboratory rodents exhibit unlearned, instinctive fear responses to advancing artificial terrestrial and aerial predators[33,34], overhead looming stimuli[35], and predator odors[36].

Here, we investigated for the first time to the best of our knowledge, whether fear conditioning readily transpires and modifies subsequent behavior of animals in a naturalistic environment. To achieve this, hunger-motivated rats searching for a food pellet in a large arena—that is, engaging in a purposive behavior as they would in nature[37]—were presented with a discrete tone CS followed by a painful US to their dorsal neck/body region by means of chronically implanted subcutaneous wires (Fig. 1a). A dorsal neck/body shock better simulates real predatory strike compared to footshock used in standard fear conditioning studies, as it is unlikely that predators direct their attacks on small prey animal's paws. In addition, in nature, bodily injuries are normally inflicted by external agents (namely, predators in animals and perpetrators in humans). Thus, other groups of rats were presented with a looming aerial predator (i.e., a lifelike great horned owl) preceded with and without a tone CS and followed by the same US (Fig. 1b–d). A single trial tone-shock, tone-owl, tone/owl-shock, and owl-shock training was employed because multiple CS-US trial-and-error (rehearsal) learning, endangering the animal to repeated bodily harm, would prove fatal in nature and is antithetical to the natural selection of

fear conditioning[23,24,32]. Later, all animals' reactions to the tone cue were examined while foraging for food in the open arena. Because the dorsal neck/body shock US has never been used before in fear research, its efficacy to support a single trial tone fear conditioning was also examined in a standard conditioning chamber.

## Results

### Baseline foraging in an ethologically-relevant environment.
Female and male rats were pseudo-randomly assigned to tone-shock (8 females, 8 males), owl-shock (8 females, 8 males), tone/owl-shock (6 females, 8 males), and tone-owl (4 females, 4 males) groups and implanted with subcutaneous wires in their dorsal neck/body (Fig. 1a–c). After recovery from surgery and habituation to the nest compartment of the arena, the hunger-motivated rats were trained to exit the nest via a computer-controlled automated gateway to procure a sizable 0.5 g food pellet placed at variable distances in the large, expanding open area of the arena (Fig. 1d, top panel). Once the animals returned to the nest for pellet consumption, the gateway closed until the next trial (3 trials/day). On the first baseline day, female rats took a significantly longer amount of time to procure the food pellet compared to male rats (Supplementary Fig. 1, Baseline day 1). This initial difference in foraging behavior likely represents heightened spatial neophobia (risk-averse to novel environments) in female rats. As rats became familiar with the foraging arena, the latency and duration measures declined across 5 baseline days comparably in both sexes, with no further statistical differences in latencies for pellet procurement. Because there were no reliable sex differences in subsequent fear conditioning dependent variables (Supplementary Fig. 2 and Supplementary Table 1), the four groups were collapsed across sexes.

### Fear conditioning in an ethologically-relevant environment.
On the training day, all rats first underwent three foraging trials with pellets fixed at the longest distance (125 cm) to confirm comparable pre-fear conditioning foraging behavior between groups (Fig. 2a, Baseline). Afterward, animals were exposed to a tone-shock, an owl-shock, a tone/owl-shock, or a tone-owl pairing in the manner shown in Fig. 1 (Supplementary Movie 1). Those rats presented with the tone CS 5-s prior to the gate opening (i.e., tone-shock, tone-owl, tone/owl-shock groups) took more time to enter the foraging arena in comparisons to owl-shock animals unexposed to the tone (Fig. 2b, Leave nest latency); this indicates that the tone was a salient cue that animals were attentive to and thus conditionable. Once in the foraging arena, all animals readily advanced toward the pellet and breached the trigger zone (25 cm from the pellet) to activate the shock, owl, or owl-shock stimuli (Fig. 2b, Trigger zone latency). In response to the shock, owl, or owl-shock, all rats promptly fled from the foraging arena to the nest (Fig. 2b, Escape latency; Fig. 2d, e, Escape speed). Figure 2c shows representative track plot examples of tone-shock, owl-shock, tone/owl-shock, and tone-owl animals successfully procuring the pellet during pre-tone baseline but not during tone conditioning. The fact that the escape latency and running speed were not significantly different between the tone-owl and other groups indicates that the looming owl-induced innate fear sans pain was just as effective in eliciting the flight UR as the painful shock or owl-shock combination. However, inspections of the escape trajectories revealed that the tone-shock and tone-owl groups tended to flee linearly to the nest, whereas the owl-shock and tone/owl-shock groups that experienced a dorsal neck/body shock 100 ms after the looming owl (mimicking realistic predatory attack) and begun their flight to the nest inclined to escape circuitously (Fig. 2f, h). This was supported by significant group differences in the escape distances (Fig. 2g) and

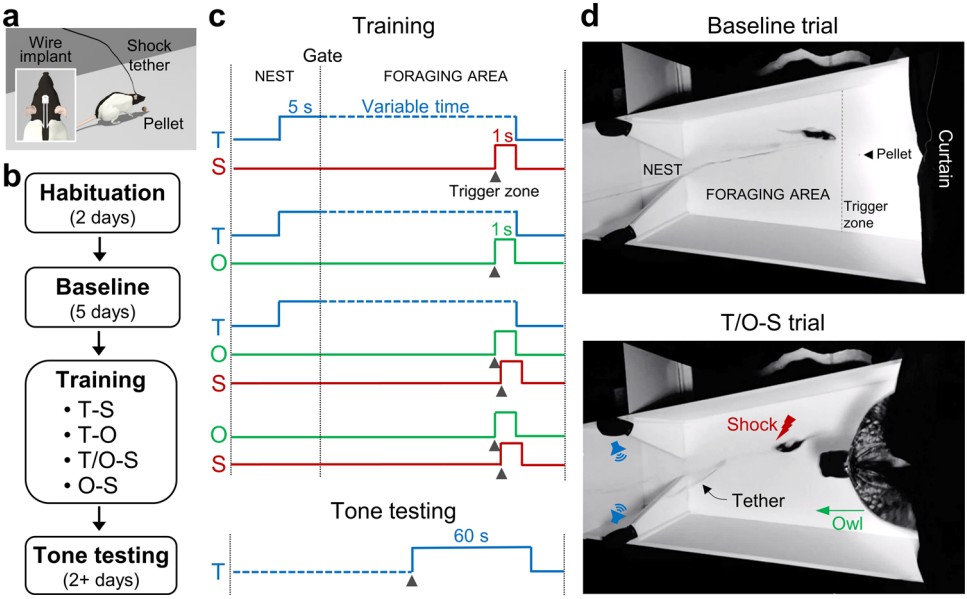

**Fig. 1 Experimental design of fear conditioning in a naturalistic setting. a** An illustration of a tethered rat foraging for a food pellet in the open arena (inset shows a headstage and placement of subcutaneous shock wires). **b** Timeline of experiment. Habituation: Rats were placed in a closed nest with dispersed food pellets for 30 min/day. Baseline: Rats were allowed to leave the nest to discover food pellets placed 25–125 cm (in 25 cm increments from the nest) in the foraging arena. Training: Animals approaching the pellet location experienced a delayed pairing of tone-shock (T-S), tone-owl (T-O), tone/owl-shock (T/O-S), or owl-shock (O-S). Tone Test: On subsequent days, all rats were placed back in the foraging arena and upon nearing the food pellet, the tone was activated. **c** Schemas of delayed pairings of stimuli. The T-S, T-O, and T/O-S (but not O-S) groups were presented with a tone 5 s before the gate opening that stayed on until the animals were within 25 cm of the food pellet, at which the tone co-terminated with the triggered shock (1 s), owl (1 s,) or owl-shock (100 ms interstimulus interval, ISI) stimuli. **d** A representative rat in the foraging arena (208 cm length × 66–120 cm expanding width × 61 cm height) during a baseline trial, where the animal successfully acquires the pellet, and during a T/O-S trial, where the animal flees from looming owl and shock into the nest (69 cm length × 58–66 cm width × 61 cm height).

variance of trajectory angles (Fig. 2i), where owl-shock and tone/owl-shock groups traveled longer distances and had higher angle variances, respectively, during their escape routes than tone-shock and tone-owl groups.

**Context (pre-tone) testing in an ethologically-relevant environment.** On the following day, animals were placed back in the nest and underwent three pre-tone baseline trials (maximum 300 s to retrieve the food pellet placed at 125 cm) to assess whether previous encounters with tone-shock, owl-shock, tone/owl-shock, and tone-owl stimuli combinations produced fear of the arena. As can be seen in Fig. 3a, the owl-shock and tone/owl-shock groups took significantly longer latencies to procure the pellet (i.e., the time from gate opening-to-return to nest with the pellet) than the tone-shock and tone-owl groups on the first day of testing. The lengthened times to enter the foraging arena exhibited by owl-shock and tone/owl-shock rats likely reflect inhibitory avoidance resulting from the previous predatory attack experience in the arena[38]. In contrast, the fact that the pre-tone test baseline latencies of tone-shock and tone-owl rats (Supplementary Fig. 3) were not reliably different from their baseline latencies from the fear conditioning day (prior to experiencing tone-shock or tone-owl) suggests that contextual fear conditioning failed to transpire in these animals despite their robust escape behavior to tone-shock and tone-owl experiences. Similar patterns of group differences, albeit lesser magnitudes, were observed on the second day of pre-tone baseline trials (Fig. 3c).

**Tone testing in an ethologically-relevant environment.** Immediately after the pre-tone baseline, all groups were subjected to three successive tone test trials (1 min apart). The owl-shock and tone/owl-shock animals continued to take longer latencies to

exit the nest compared to tone-shock and tone-owl animals (Fig. 3b, Leave nest latency). Once in the foraging arena, the tone/owl-shock group's latency to approach 25 cm from the pellet to trigger the tone were marginally but reliably longer than those of tone-shock and tone-owl groups, but not the owl-shock group (Fig. 3b, Trigger zone latency). Upon the activation of tone (60 s continuous), the majority of owl-shock and tone/owl-shock animals promptly fled to the nest (Supplementary Movie 2), thereby significantly increasing the latency to procure the pellet (60 s = unsuccessful), whereas the tone-shock and tone-owl animals were largely unaffected by the tone and readily procured the pellet (Fig. 3b, Procure pellet latency). No freezing (as measured by the ANY-maze tracking software with a 2 s threshold) was detected in the foraging arena during the tone presentations. The second day of tone testing yielded similar patterns of group differences (Fig. 3d). Figure 3e shows individual track plots from all animals with the initial number of trial(s) necessitated for successful foraging. Further analyses across tone testing days (3 trials/day) showed that the overall success rates of procuring the pellet were significantly lower in owl-shock and tone/owl-shock groups compared to tone-shock and tone-owl groups (Fig. 3f), and that owl-shock and tone/owl-shock animals required extended trials to reliably obtain the pellet (Fig. 3g). Because the temporal interval between the CS and US is well known to be crucial in various types of Pavlovian conditioning, including fear conditioning[39], we examined whether tone fear conditioning transpired in a specific (optimal) range of interstimulus intervals (ISI) but was masked by non-optimal ISIs. We found no significant correlation between the ISIs and the magnitudes of tone-induced suppression of pellet procurement in tone-shock animals, indicating that tone fear conditioning failed to materialize across varying ISIs of delay conditioning (Fig. 3h). Conversely, in the tone/owl-shock

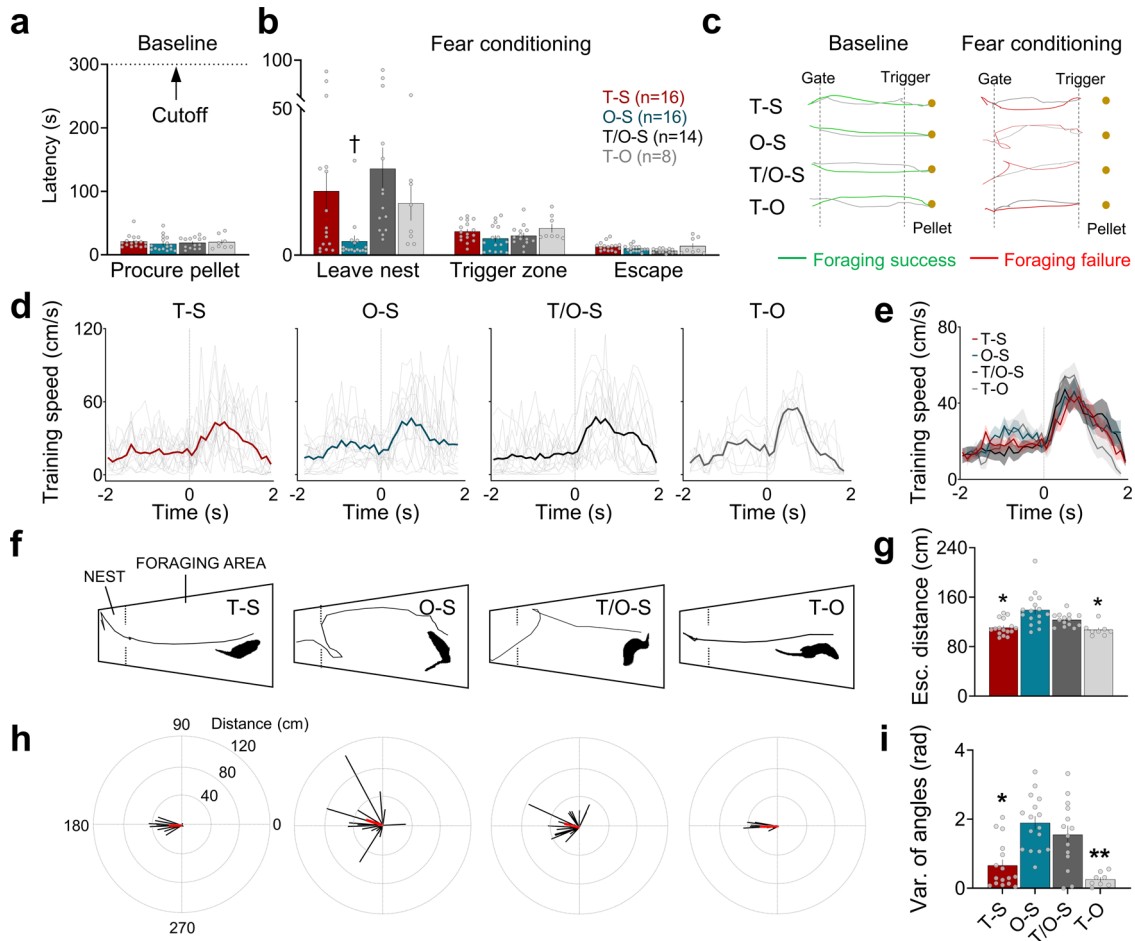

**Fig. 2 Foraging and escape behaviors during fear conditioning. a** Pre-conditioning baseline latencies (mean ± SEM) to procure food pellets in the foraging arena were equivalent between T-S (red), O-S (blue), T/O-S (dark gray), and T-O (light gray) groups (Kruskal–Wallis, H = 2.694, $p$ = 0.441). **b** During fear conditioning, the T-S, T/O-S, and T-O groups exposed to the tone 5 s before the gate opening had significantly longer latencies to leave the nest than the O-S group (left panel, Kruskal–Wallis, H = 18.6, $p$ < 0.001; pairwise comparisons, $p$ = 0.008 for T-S vs. O-S, $p$ = 0.011 for O-S vs. T-O, $p$ < 0.001 for O-S vs. T/O-S, $p$ = 0.69 for T-S vs. T-O, $p$ = 0.631 for T-S vs. T/O-S, $p$ = 0.343 for T/O-S vs. T-O). Once outside the nest, however, the latency to breach the trigger zone, enroute to the pellet, was not reliably different among the groups (Kruskal–Wallis, H = 7.453, $p$ = 0.059). In response to the triggered shock, owl or owl-shock, all groups showed similar escape-to-nest latencies (Kruskal–Wallis, H = 6.141, $p$ = 0.105). **c** Representative track plot examples from T-S, O-S, T/O-S, and T-O animals during the baseline, when animals successfully procured the pellet, and during the fear conditioning, when the same animals fled from shock, owl or owl-shock stimuli and thus unable to attain the pellet. **d** Mean instantaneous speed (±SEM) of each group 2 s before and after the shock, owl or owl-shock onset ($t = 0$). Thin, gray lines represent individual animal data. **e** All groups showed comparable escape speed to the shock, owl, and owl-shock stimuli (Kruskal–Wallis, H = 0.901, $p$ = 0.825). **f** Representative track plots showing escape paths of T-S, O-S, T/O-S, and T-O animals. The inset silhouette images show that the T-S and T-O animals were facing forward at the time of the shock or owl stimulus whereas the O-S and T/O-S animals were turning back at the time of the shock stimulus because of the 100 ms owl-shock interstimulus interval. **g** Mean escape distance (±SEM) from the trigger zone to the nest. The O-S and T/O-S groups traveled longer distances to escape compared to the T-S and T-O groups (Kruskal–Wallis, H = 21.98, $p$ < 0.001; pairwise comparisons, $p$ = 0.014 for T-S vs. T/O-S, $p$ = 0.008 for T/O-S vs T-O, $p$ = 0.001 for T-S vs. O-S, $p$ = 0.001 for O-S vs T-O). **h** Representative vector plots of each group showing variabilities in their escape paths. **i** Mean variance (±SEM) of escape trajectory angles (radian) from the trigger zone to the nest. The O-S and T/O-S groups had greater variance in their escape trajectories when fleeing back to the nest (Kruskal–Wallis, H = 22.37, $p$ < 0.001; pairwise comparisons, $p$ = 0.022 for T-S vs. T/O-S, $p$ = 0.003 for T/O-S vs T-O, $p$ = 0.002 for T-S vs. O-S, $p$ < 0.001 for O-S vs T-O) († compared to T-S, T/O-S, and T-O; * compared to O-S and T/O-S, $p$ < 0.05, **$p$ < 0.01, ***$p$ < 0.001; # compared to T/O-S, $p$ < 0.05, ##$p$ < 0.01).

animals, the tone-induced suppression of pellet procurement was uniformly observed across different ISIs, suggesting that the observed fear in these animals may not necessarily reflect Pavlovian conditioning (Fig. 3h). Moreover, there were no significant differences between the owl-shock and tone/owl-shock animals on any of the measures (Supplementary Fig. 4). In particular, the latencies to pellet procurement during the 3rd and 4th tone tests were not correlated with the tone-shock intervals during training even though there was more variability within the tone/owl-shock group's procurement times. The instantaneous speeds to the tone and angle trajectories of escape

were also similar between the owl-shock and tone/owl-shock groups (Supplementary Fig. 5). These results suggest that there seems not to be an additional influence of Pavlovian memory in the tone/owl-shock group. The key results of delayed tone-shock paired animals failing to show conditional tone fear and contextual fear suggest that standard fear conditioning does not readily occur in a naturalistic environment. Instead, the finding of owl-shock animals displaying robust fear to a novel tone, which the animals never heard before, suggests that non-associative processes play a crucial role in protecting animals in the real world.

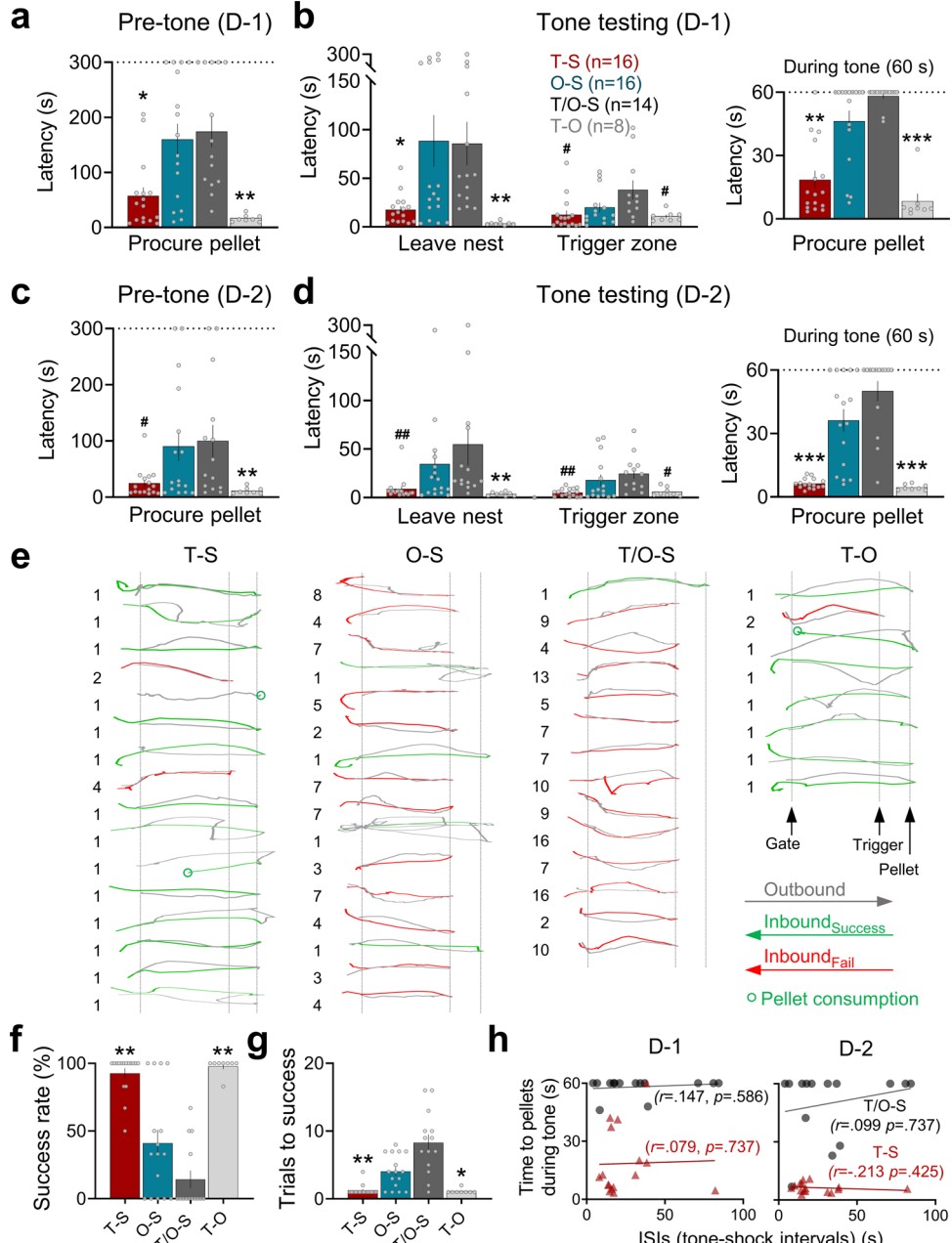

**Tone testing inside the nest**. It is possible that tone fear conditioning transpired in the T-S group (Fig. 3) but conditional fear behavior was not observed because the tone CS was presented while the animals were in the large foraging arena. To address this, 8 experimentally naive rats (4 females, 4 males) underwent the identical surgical, food restriction, habituation, baseline foraging, and tone-shock conditioning procedures as described above (Fig. 1a, b). After three pre-tone baseline trials, the tone CS was activated while the animals were inside the nest with the gateway closed. During the 60 s of continuous tone, none of the animals exhibited reliable freezing behavior (Supplementary Fig. 6a, left). When the gateway opened while the tone CS remained on, all rats readily entered the foraging arena and procured the pellet (Supplementary Fig. 6a, right). The animals then underwent 2 additional tone-shock pairings and were retested in the same manner. Even after a total of 3 tone-shock pairings, the tone CS failed to elicit freezing inside the nest with the gateway closed and inhibit/delay the latency to foraging when

the gateway opened (Supplementary Fig. 6b). These results further suggest that no associative learning to the tone CS was acquired in our naturalistic environment.

**Fear conditioning in a standard chamber**. To determine whether the absence of tone fear conditioning in a naturalistic environment (Fig. 3) was due to rats receiving subdermal pain to their dorsal neck/body region, as opposed to dermal pain to their paws in standard fear conditioning, 8 other experimentally naive rats (4 females and 4 males) that underwent the same aforementioned subcutaneous wire implant surgery, food restriction, habituation, and baseline foraging procedures were presented with a tone CS and dorsal neck/body shock US pairing in a standard conditioning chamber (Fig. 4a). An additional 8 experimentally naive rats (4 females and 4 males), except for being *ad lib*-fed akin to most fear conditioning studies (e.g., refs. [38–40]), underwent the same tone CS-dorsal neck/body US pairing. The fixed CS duration (24.1 s) employed was based

**Fig. 3 Foraging and escape behaviors during tone testing. a** The mean latency (±SEM) to procure the pellet during the pre-tone baseline trials on testing day 1 (D-1). Both O-S and T/O-S groups took significantly longer times to exit (gate opening, $t = 0$) and return to the nest with the pellet than T-S and T-O groups (Kruskal–Wallis, H = 20.518, $p < 0.001$; pairwise comparisons, P = 0.003 for T-S vs. T/O-S, $p < 0.001$ for T/O-S vs. T-O, $p = 0.013$ for T-S vs. O-S, $p < 0.001$ for O-S vs. T-O). **b** The times (mean ± SEM) to leave nest and reach trigger zone on day 1 tone test trials. Both O-S and T/O-S groups had longer latencies to leave nest (Kruskal–Wallis, H = 27.071, $p < 0.001$; pairwise comparisons, $p = 0.003$ for T-S vs. T/O-S, $p < 0.001$ for T/O-S vs. T-O, $p = 0.044$ for T-S vs. O-S, $p < 0.001$ for O-S vs. T-O. Once outside the nest, the T/O-S group took longer time to reach the trigger zone than the T-S and T-O (Kruskal–Wallis, H = 9.153, $p = 0.027$; pairwise comparisons, $p = 0.019$ for T-S vs. T/O-S, $p = 0.042$ for T/O-S vs. T-O). During the tone test, the latencies to procure the pellet within the 60 s allotted time were significantly longer in O-S and T/O-S animals compared to T-S and T-O animals (Kruskal–Wallis, H = 34.428, $p < 0.001$; pairwise comparisons, $p < 0.001$ for T-S vs. T/O-S, $p < 0.001$ for T/O-S vs. T-O, $p = 0.002$ for T-S vs. O-S, $p < 0.001$ for O-S vs. T-O). **c** The mean latency (±SEM) to procure the pellet during the pre-tone baseline trials on testing day 2 (D-2). O-S and T/O-S groups continued to have longer latencies to exit (gate opening, $t = 0$) and return to the nest with the pellet than T-S and T-O groups (Kruskal–Wallis, H = 12.47, $p = 0.006$; pairwise comparisons, $p = 0.022$ for T-S vs. T/O-S, $p = 0.002$ for T/O-S vs. T-O, P = 0.009 for O-S vs. T-O). **d** The times (mean ± SEM) to leave nest and reach trigger zone on day 2 tone test trials. There were group differences in the latencies to leave nest (Kruskal–Wallis, H = 21.505, $p < 0.001$; pairwise comparisons, $p = 0.001$ for T-S vs. T/O-S, $p < 0.001$ for T/O-S vs. T-O, $p = 0.002$ for O-S vs. T-O). Once outside the nest, there were group differences in the latencies to reach the trigger zone (Kruskal–Wallis, H = 21.531, $p < 0.001$; pairwise comparisons, $p < 0.001$ for T-S vs. T/O-S, $p < 0.001$ for T/O-S vs. T-O, $p = 0.037$ for O-S vs. T-O). During the tone test, the latencies to procure the pellet within the 60 s allotted time were significantly longer in O-S and T/O-S animals compared to T-S and T-O animals (Kruskal–Wallis, H = 37.223, $p < 0.001$; pairwise comparisons, $p < 0.001$ for T-S vs. T/O-S, $p < 0.001$ for T/O-S vs. T-O, $p < 0.001$ for T-S vs. O-S, $p < 0.001$ for O-S vs. T-O). **e** Individual track plots from all animals from each group displaying the XY trajectory coordinates each rat took during the first tone exposure. The parenthesized numbers next to plots represent the trial(s) needed for successful foraging. **f** The overall success rates of procuring the pellet on the first testing day were significantly lower in the O-S and T/O-S groups compared to the T-S and T-O groups (Kruskal–Wallis, H = 32.299, $p < 0.001$; pairwise comparisons, $p < 0.001$ for T-S vs. T/O-S, $p < 0.001$ for T/O-S vs. T-O, $p = 0.001$ for T-S vs. O-S, $p = 0.003$ for O-S vs. T-O). **g** The O-S and T/O-S animals required extended trials to obtain the pellet (Kruskal–Wallis, H = = 32.004, $p < 0.001$; pairwise comparisons, $p < 0.001$ for T-S vs. T/O-S, $p < 0.001$ for T/O-S vs. T-O, $p = 0.002$ for T-S vs. O-S, $p = 0.011$ for O-S vs. T-O). **h** In T-S and T/O-S animals, there were no reliable correlations (Spearman's correlation coefficient) between the tone-induced suppression of pellet procurement (an index of fear) and the temporal intervals (i.e., ISIs) between tone CS onset and shock US onset in neither testing day 1 nor 2 (* compared to both O-S and T/O-S, $p < 0.05$, **$p < 0.01$, ***$p < 0.001$; # compared to T/O-S, $p < 0.05$, $p < 0.01$).

on the mean CS duration of tone-shock animals in the naturalistic fear conditioning experiment (Fig. 3h). Following the CS-US pairing, both restricted-food and *ad lib*-food animals exhibited reliable postshock freezing (fear conditioning day 1; Fig. 4b, e and Supplementary Fig. 7a, d) and tone CS-elicited freezing in a contextually-altered chamber (tone testing day 2; Fig. 4c, d, f, g and Supplementary Fig. 7b, c, e, f). There were no reliable group differences between restricted-food and *ad lib*-food animals in postshock freezing (Fig. 4b, 47.76 ± 4.24% vs. Fig. 4e, 52.1 ± 7.12%; independent *t* test, $t(14) = -0.365$, $p = 0.721$) and tone CS-elicited freezing (Fig. 4d, 52.44 ± 4.66% vs. Fig. 4g, 46.04 ± 11.8%; independent *t* test, $t(14) = 0.504$, $p = 0.622$). The fact that fear conditioning transpired with a single tone-shock pairing in a standard chamber comparably in restricted-food animals and *ad lib*-food animals suggests that the absence of conditioned tone-elicited fear in a naturalistic environment is unlikely due to attributes of tone CS and dorsal neck/body shock US (as opposed to a footshock) or due to sustained hunger motivation.

## Discussion

It is generally believed (though never validated) that there is behavioral continuity of Pavlovian fear conditioning from the laboratory to real-life situations, and thus understanding the mechanisms of fear conditioning will have clinical relevance. The present study directly investigated whether fear conditioning readily occurs in naturalistic situations that animals are likely to encounter in their habitats. Standard fear conditioning in rodents takes place in small experimental chambers, and several studies have shown that a single tone CS-footshock US pairing (i.e., delay fear conditioning) reliably produces conditioned freezing in rats and conditioned tachycardia/freezing in mice[40]. One-trial delay tone fear conditioning has also been demonstrated in human subjects using a loud white noise US and assessing conditioned skin conductance response[41]. However, in the present study, where rats are exhibiting a purposive foraging behavior[37] in a

large arena, a delayed pairing of tone CS and dorsal neck/body shock US (tone-shock group) produced virtually no evidence of auditory (and contextual) fear conditioning across a range of CS durations (i.e., ISIs). A similar pairing of tone CS and looming owl (tone-owl group) also failed to produce auditory fear conditioning despite the owl US evoking robust fleeing UR. In contrast, foraging rats that experienced a looming owl and shock pairing (owl-shock group) later exhibited robust fear (escape) behavior to a novel tone presentation. In the tone/owl-shock animals, the escape behavior was uniformly observed across different ISIs, suggesting that the observed fear to the tone CS in this group may also not be a Pavlovian response. These findings then point to a nonassociative process rather than associative tone fear memory, as playing a vital function in risky (i.e., predatory attack) situations that animals encounter in nature. Specifically, the owl-shock condition, where a novel tone prompted similar fleeing behavior caused by the owl-shock experience the previous day, may represent pseudo-conditioning, which refers to UR-like behavior emerging to a novel stimulus after mere exposure to a biologically significant US[42,43]. The observed contextual fear and subsequent fleeing to the novel tone in owl-shock (as well as tone/owl-shock) animals is also consistent with the finding that pseudo-conditioning transpires from conditioning of the context[44].

The tone CS (3 kHz, 80 dB, ranging 9–86.6 s) and subcutaneous dorsal neck/body shock US (2.5 mA, 1 s) employed in the present study were effective in eliciting orienting and fleeing responses, respectively, and were presented to animals in the manner (i.e., a delay conditioning) that satisfied the stimuli saliency, intensity, surprising, and temporal contiguity requirements for conditioning[45–47]. Indeed, the same dorsal neck/body shock served as an effective US to generate one-trial tone fear conditioning in a standard (small) conditioning chamber in both restricted-food and *ad lib*-food animals. Then, what can account for one-trial auditory fear conditioning, demonstrated in standard Pavlovian paradigms in rats, mice, and humans[38,40,41,48], not emerging in animals that left the safe nest to forage for food in an

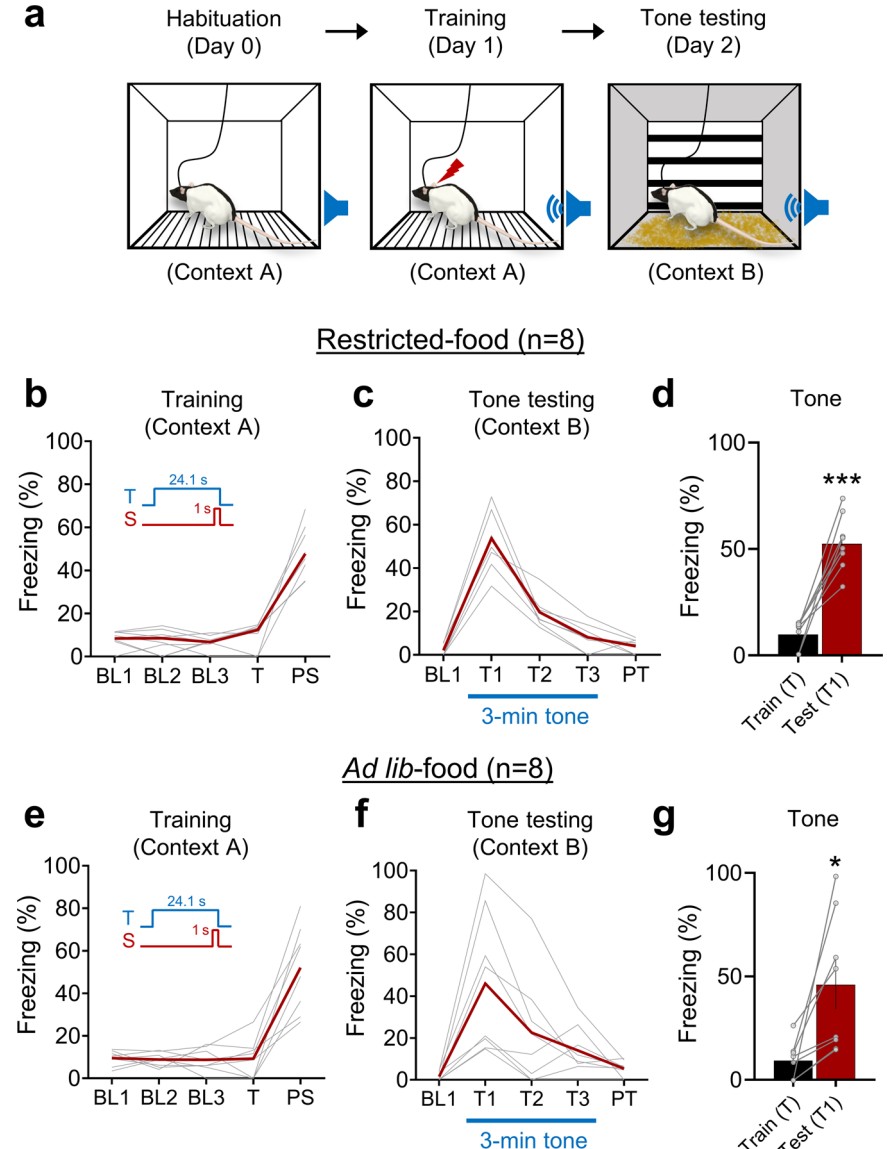

**Fig. 4 Auditory fear conditioning in a standard experimental chamber. a** Illustrations of a rat implanted with wires subcutaneously in the dorsal neck/body region undergoing successive days of habituation (10 min tethered, conditioning chamber), training (a single tone CS-shock US pairing), and tone testing (context shift). (restricted-food) **b** Mean (crimson line) and individual (gray lines) percent freezing data from 8 rats (4 females, 4 males) during training in context A: 3 min baseline (BL1, BL2, BL3); 23.1 s epoch of tone (T); 1 min postshock (PS). **c** Mean and individual percent freezing data during tone testing in context B: 1 min baseline (BL1); 3 min tone (T1, T2, T3); 1 min post-tone (PT). **d** Mean ± SEM (bar) and individual (dots) percent freezing to tone CS before (Train, T) and after (Test, T1) undergoing auditory fear conditioning (paired $t$ test; $t(7) = -7.319$, $p < 0.001$). (ad lib-food) **e** Mean (crimson line) and individual (gray lines) percent freezing data from 8 rats (4 females, 4 males) during training in context A: 3 min baseline (BL1, BL2, BL3); 23.1 s epoch of tone (T); 1 min postshock (PS). **f** Mean and individual percent freezing data during tone testing in context B: 1 min baseline (BL1); 3 min tone (T1, T2, T3); 1 min post-tone (PT). **g** Mean ± SEM (bar) and individual (dots) percent freezing to tone CS before (Train, T) and after (Test, T1) undergoing auditory fear conditioning (paired $t$ test; $t(7) = -3.188$, $p = 0.015$). *$p < 0.05$, ***$p < 0.001$.

open arena? It may well be that rats are not biologically predisposed to associate discrete CS and US in natural (complex) environments where amalgamation of hunger-driven, fear-driven, and exploration-driven motivated behaviors are freely expressed. Indeed, in real-life, only a small minority of people experiencing trauma develop posttraumatic stress disorder (PTSD) and even with re-exposure to the same trauma there is low incidence of PTSD[49,50]. In contrast, standard experimental chambers may be conducive to fear conditioning because they are simple and limit the repertoire of behavior[31], effectively bypassing a "biological boundary" that prioritizes less costly defensive responses over trial-and-error learning mechanisms. The absence of one-trial fear conditioning in a naturalistic setting may be analogous to

"The Rat Park Experiment," where rats housed in an enriched environment with plants, trees, and social interaction resist drug addiction behavior evident in standard cage-housed rats[51,52]. Animals tested in naturalistic paradigms are given choices that do not force their behaviors into dichotomies (i.e., freezing or no freezing; drug craving or no drug craving). Allowing for an expanded behavioral repertoire, while more difficult to study, may thus yield a greater understanding of behaviors and their underlying brain mechanisms.

It should also be noted that fear encounters in real life generally occur in the presence of external agents or forms (i.e., predators/conspecifics in animals and assailants/combatants in humans), which is virtually nonexistent in standard Pavlovian fear

conditioning paradigms. Thus, the effects of a discernable entity in associative fear learning have never been investigated. By simulating a realistic life-threatening situation, i.e., a looming aerial predator that instinctively elicited flight behavior followed by somatic pain, we found that rats engaged in purposive behavior likely utilize nonassociative pseudo-conditioning as their primary defensive mechanism. The fact that the owl-shock and tone/owl-shock animals exhibited relatively nonlinear, erratic escape trajectories to the nest compared to linear escape trajectories in tone-shock animals (Fig. 2f–i) suggests the intriguing possibility that the same dorsal neck/body shock US may be interpreted as a life-or-death (panic) situation in the presence of an external threat agent versus a mere startling (nociceptive) situation in the absence of an external threat agent. The erratic flight behavior in the presence of a looming owl may represent the penultimate stage of circa-strike, or "life-or-death," behavior within the "predatory imminence continuum" theory[53]. Functionally, a 'sensitized' fear system may intensify avoidance behavior, which in turn effectively transposes novel, neutral cues into "false positives" to prioritize survival in natural environments[32]. In other words, nonassociative process-based overestimation/generalization of danger may be a more prudent course for survival than associative process-based specific prediction of danger.

The present owl-shock-like procedure can perhaps be introduced in standard conditioning chambers outfitted with overhead monitors to produce two-dimensional (2D) looming stimuli (e.g., a rapidly expanding black disc) that can evoke freezing if the animal is distant from an enclosed shelter or fleeing if the animal is nearby an enclosed shelter[35]. In doing so, footshock can be delivered as the animals are either freezing or fleeing to the 2D looming disc to potentially investigate, for example, nonassociative aspects of fear mechanisms and whether the same predatory strike evokes differential defensive responses in animals are engaged in central amygdala-mediated freezing vs. basolateral amygdala-mediated active avoidance[54,55]. Incorporating external agents of danger into standard fear conditioning paradigms may lead to more realistic translational findings. However, it must also be considered that fear behaviors observed in an ecologically-relevant environment, where instinctive activities are unconstrained, might not be similarly observed in a standard operant chamber, where conditioning plays disproportionately dominant role over instinctive fear, and vice versa[31].

Some caveats, however, must be considered in the present naturalistic study of fear conditioning. First, although neither the tone-shock group nor the tone-owl group showed overt manifestations of fear conditioning to the tone (as measured by fleeing or freezing in the arena) that prevented a successful procurement of food, the possibility of physiological (e.g., cardiovascular, respiratory) indices of fear[56] cannot be excluded in these animals. If so, the *presence* of tone-elicited fleeing and foraging termination behaviors in owl-shock and tone/owl-shock animals versus the *absence* of tone-elicited fleeing and foraging termination behaviors in tone-shock and tone-owl animals may reflect differences in the magnitude (rather than presence-absence) of fear conditioning. Second, the erratic escape trajectory behavior exhibited by owl-shock and tone/owl-shock animals may be indicative of rapid associative processes at work[57]. For example, the immediate-shock (and delayed shock-context shift) deficits in freezing, e.g., refs. [58,59], provide compelling evidence that post-shock freezing is not a UR but rather a CR to the contextual representation CS that rapidly became associated with the foot-shock US. In a similar vein then the erratic escape CR topography in owl-shock and tone/owl-shock animals might represent a shift in 'functional CR topography'[47] resulting from the rapid association between some salient features of the owl and the dorsal neck/body shock. A rapid owl-shock association nevertheless cannot explain the owl-shock animals' subsequent fleeing behavior to a novel tone (in the absence of owl), which likely reflects nonassociative fear. Third, there are obvious procedural differences between standard fear conditioning versus naturalistic fear conditioning. In the former paradigm, typically ad libitum-fed animals are placed in an experimental chamber for a *fixed time* before receiving a CS-US pairing (irrespective of their ongoing behavior). Thus, the CS duration and ISI are constant across subjects. In our study, hunger-motivated rats searching for food must navigate to a *fixed location* in a large arena before experiencing a CS-US pairing (instrumental- or response-contingent). Because animals approach the US trigger zone at different latencies, the CS duration and ISI are variable across subjects. A more pertinent question is whether "procedurally pure" laboratory Pavlovian fear conditioning can possibly occur in real-world settings, where behaviors of animals and humans are largely purposive/goal-oriented[37]. Indeed, Bouton[43] articulated that, "Outside the laboratory, stimulus [Pavlovian] learning and response [Instrumental] learning are almost inseparable." Last, tone fear conditioning might not have transpired in our foraging apparatus because the shock-induced pain was targeted to the dorsal neck/body region. As stated before, this is unlikely given that the same dorsal neck/body shock US effectively supported single trial tone fear conditioning in a standard conditioning chamber. Though predators would not direct their attacks underneath the paws of small prey animals, the possibility of a footshock US supporting tone fear conditioning in the foraging apparatus, however, cannot be excluded.

Clark Hull[60] has posited that Pavlovian fear conditioning offers biological utility by circumventing a "bad biological economy" of defense reaction always necessitating injury. This prevailing view that ascribes preeminent importance of fear conditioning as the primary defensive mechanism is likely to be a theoretical simplification and provides an incomplete picture of fear, as its function in a natural environment may be rather limited (i.e., lacks face validity). It may well be possible to produce fear conditioning in naturalistic settings with further CS-US trials, varying the CS and US intensity/duration or applying footshock but then this too would be a bad biological economy as such specific parameter-dependent learning would dramatically reduce biological fitness. It is also important to recognize inconsistencies in the literatures, such as clinical studies that have reported that patients with anxiety disorders, such as phobias, have trouble recalling the particular pairing of the fear event with its aversive consequences[61,62]. The increased utilization of naturalistic fear paradigms that simulate dangers that animals and humans encounter in real life will enable us to clarify, update, and revise fear concepts derived largely from fear conditioning studies and in doing so facilitate future progress in the treatment of fear disorders.

## Methods

**Subjects**. Eighty-six Long-Evans rats (3–4 months old; 44 females and 42 males, RRID:RGD_2308852), purchased from Charles-Rivers Laboratories, were initially pair-housed by sex for 5–7 days of acclimatization in a climate-controlled vivarium (accredited by the Association for Assessment and Accreditation of Laboratory Animal Care), with a reversed 12-h light/dark cycle (lights on at 7 PM). After undergoing subcutaneous wire implant surgery (described below), all animals were individually housed. Of 86 rats, 78 were placed on a standard restricted-food schedule with *ad lib* access to water to gradually reach and maintain ~85% normal body weight while the remaining 8 had *ad lib* access to both food and water. All experiments were performed during the dark phase of the cycle in strict compliance with the University of Washington Institutional Animal Care and Use Committee guidelines.

**Surgery**. Under isoflurane anesthesia, rats were mounted on a stereotaxic instrument (Kopf), and two Teflon-coated stainless-steel wires (0.0003 inch bare,

0.0045 inch coated; A-M Systems, Everett, WA) were inserted in the dorsal neck/back region of body. The wire tips were exposed (~1 cm), bent to a V-shape, and hooked to subcutaneous tissue[39]. The other ends of the wires were affixed to a headstage (Plastics One, MS303-120), which was then cemented to the animal's skull embedded with 6 anchoring screws. While still under anesthesia, animals were connected to a shock-apparatus and given a mild shock to observe muscle twitching; 6 rats that showed no reaction to shock were removed from the experiment. Animals were given 4 days of postoperative recovery and were adapted to handling for 5 days before nest habituation.

**Foraging apparatus and stimuli**. A custom-built foraging arena consisted of a nest (69 cm length × 58–66 cm width × 61 cm height) that opened via an automated sliding gate to reveal a large, expanded foraging area (208 cm length × 66–120 cm width × 61 cm height) where 0.5 g food pellets (grain-based; F0171, Bio-Serv) were placed at variable locations (Fig. 1a). The testing room was kept under red light (11 lux foraging area, 2 lux nest area) with constant white noise (72 dB) playing in the background. Prior to placing each animal, the arena was wiped with 70% ethanol. The ANY-maze software and Ami interface system (Stoelting) connected to a PC automatically tracked the animal's position in the arena, via a ceiling mounted camera, and triggered the tone, shock, and aerial predator stimuli: (i) 3 kHz, 80 dB tone CS (measured from the trigger location; 81 dB within the nest area) was produced using ANY-maze (Stoelting) and presented through two speakers mounted on the nest-foraging border; (ii) 1 s, 2.5 mA shock US was delivered to the animal's dorsal neck/back region via a headstage tethered to a stimulus-isolator (Bak); (iii) A life-like model owl[34], mounted onto a 92 cm pneumatic air cylinder (Bimba) at the opposite end of the foraging arena and hidden behind a black curtain, plunged downward towards the rat (46 cm/s), then retracted back to it starting position.

**Behavioral procedure for naturalistic fear conditioning**. A total of 62 rats (32 females and 30 males, all restricted-food) were used to investigate fear conditioning in an ecologically-relevant environment. Upon reaching and maintaining 85% normal body weight, animals were transported to the experimental room and underwent series of habituation, baseline, fear conditioning, and testing sessions.

*Habituation days*. Animals were placed in the nest scattered with 20 food pellets (0.5 g, grain-based, Bio-Serv) for 30 min/day for 2 consecutive days to acclimatize and associate the nest with food consumption.

*Baseline days*. After 1 min in the nest sans food pellets, the gate opened, and the animal was allowed to explore the large foraging arena and find a pellet placed 25 cm away from the nest (first trial). As soon as the animal took the sizeable 0.5 g pellet back to the nest, the gate closed. Once the animal finished eating, the second trial with the pellet placed 50 cm and then the third trial with the pellet placed 75 cm commenced in the same manner. Animals underwent 3–5 consecutive baseline days, with the pellet distances gradually extending to 75, 100, and 125 cm, and they were also accustomed to tethering beginning on baseline day 3 onward.

*Fear conditioning day*. Rats, pseudo-randomly assigned into tone-shock, tone-owl, tone/owl-shock, and owl-shock groups (Fig. 1), underwent 3 baseline trials with the pellet placed at 125 cm from the nest. On the 4th trial, the tone-shock, tone-owl, and tone/owl-shock animals were exposed to a tone CS that came on 5 s before the gate opened and remained on until they reached the trigger zone (25 cm to the pellet). For tone-shock and tone-owl animals, the tone co-terminated with the shock US and the owl looming, respectively. For tone/owl-shock animals, the shock occurred 0.1 s after the owl looming and co-terminated with the tone. Two animals in the tone/owl-shock group were excluded because they failed to leave the nest within 2 min. The owl-shock animals were subjected to the same owl looming-shock pairing (as the tone/owl-shock animals) but in the absence of tone. All rats fled to the nest in reaction to the shock and/or looming owl, at which time the gate was closed. After 1 min in the nest, the animals were placed back into their homecage.

*Testing days*. All rats underwent 3 baseline trials (a maximum of 300 s to retrieve the pellet) to assess whether shock and/or looming owl encounter the previous day resulted in the fear of the arena (i.e., contextual fear). Afterward, animals were presented with the tone cue when they approached the trigger zone (25 cm to the pellet). The tone played continuously for 60 s, after which the tone test trial ended. Animals underwent 3 tone tests daily until they successfully attained the pellet (i.e., fear extinction).

**Behavioral procedure for tone testing inside the nest**. Another tone-shock group of 8 rats (4 female, 4 males; restricted-food) underwent the same Habituation, Baseline, and Fear conditioning procedures described above. On the testing day, after 3 baseline trials (a maximum of 300 s to retrieve the pellet), while the animals were inside the nest with the gateway closed, the tone CS was activated continuously, and freezing behavior was measured for 60 s (freezing analysis described below). Then, while the tone remained on, the gateway opened to assess the latency to procure the pellet. Once the animals returned to the nest with the pellet, the tone CS terminated.

**Behavioral procedure for standard fear conditioning**. The remaining 16 rats (4 females and 4 males, restricted-food; 4 females and 4 males, *ad lib*-food) were subjected to one-trial tone fear conditioning in a standard conditioning chamber[38] instead of an ecologically-relevant foraging arena. The restricted-food animals underwent the aforementioned Habituation and Baseline procedures prior to standard fear conditioning, whereas the *ad lib*-food animals proceeded to standard fear conditioning directly (similar to most fear conditioning studies). A day prior to fear conditioning (day 0), both restricted-food and *ad lib*-food animals were tethered and placed in an experimental chamber for 10 mi of pre-exposure. Fear conditioning (day 1) commenced after 3 min of baseline in the chamber by exposing animals to 24.1 s tone CS (3 kHz, 80 dB) that co-terminated with 1 s dorsal neck/body shock (2.5 mA). The 24.1 s tone was based on the mean tone CS duration from the tone-shock group of the naturalistic fear conditioning experiment. Postshock freezing was assessed for 1 min before animals were removed from the conditioning chamber. For tone test (day 2), animals were placed in a novel chamber that differed in terms of the wall pattern, floor texture, background light, and smell[63]. After 1 min of baseline, the tone was presented for 3 min to assess CS-evoked freezing response, and the animals were left in the chamber for an additional minute before being placed back in their homecage. Freezing was again quantified using ANY-maze (Stoelting) tracking software with the freezing threshold set to 2 s.

**Statistics and reproducibility**. Statistical analyses were performed using SPSS (IBM, version 19) and R (The R Foundation, version 3.5.3). Body tracking positions were obtained using Deep Lab Cut[64] and analyzed using a self-written script in Python (Python Software Foundation). Animal sample sizes were determined using a power analysis performed by G*Power (G*Power, version 3.0.1, Franz Faul; power = 0.95, alpha = 0.05, effect size = 0.5, two-tailed). A Levene's test for normality showed significance for the data, thus nonparametric tests were used for analyses. Because there were no significant sex differences in any stages of the experiment after the first day of baseline (Supplementary Fig. 1 and Supplementary Table 1), data from females and males were pooled together for all analyses (Supplementary Fig. 2). Statistical significance was set at $P < 0.05$. Graphs were made using GraphPad Prism (version 8).

For the analyses of escape trajectories (Fig. 2h, i), the coordinate data of each rat in the foraging arena taken at a frequency of 10 Hz was used to obtain the change in position vectors between each time point (black) and an overall change in position vector (red). To obtain the individual change in position vectors, we used Python and the Numpy, Pandas, and Matplotlib packages to calculate the changes in x and y position between coordinates. With each change in x and y positions, we were able to calculate the magnitude of the distance traveled and the angle of travel using an inverse tangent function. The resultant vector representing the average change in position vector was determined by taking the average change in x position and average change in y position to calculate an overall magnitude and angle. The (population) variance and standard deviation of the angles of the change in position vectors were obtained using Numpy.

**Reporting summary**. Further information on research design is available in the Nature Research Reporting Summary linked to this article.

## Data availability
The data that support the findings of this study and the relevant analysis code are available from the Dryad data repository. https://doi.org/10.5061/dryad.76hdr7sxk.

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

## Acknowledgements

We thank Lori A. Zoellner for valuable comments on the manuscript, and Heather Wu for assistance in the experiment. This study was supported by National Institutes of Health Grant MH099073 (to J.J.K.).

## Author contributions

P.R.Z., E.J.K., and J.J.K. designed the study; P.R.Z. and B.E.L. collected the data; P.R.Z., B.P.S., and A.S. conducted the statistical analyses; P.R.Z., B.P.S., E.J.K., and J.J.K. wrote the manuscript. J.J.K. supervised all aspects of the study.

## Competing interests

The authors declare no competing interests.
