## [Peer Review File · Communications Biology]

Reviewers' comments:

Reviewer #1 (Remarks to the Author):

In this manuscript, the authors examine whether single-trial auditory fear conditioning and contextual conditioning occurs in rats in a naturalistic foraging setting, and they also examine what happens to these forms of fear learning when a fear-inducing predatory stimulus precedes the shock. The relevance of standard laboratory fear conditioning to real-world fear learning is rarely addressed, but a very important question.

Male and female rats were trained to leave a "nest" area to retrieve food pellets at the end of a long hallway extending from the next area. They were then subjected to one of four types of stimulus pairing: tone-shock, tone-owl-shock, owl-shock, or tone-owl. The following day, contextual fear was assessed by returning animals to the apparatus and measuring time to retrieve pellets. Auditory fear was assessed by returning animals to the apparatus, playing a 60-s continuous tone, and measuring time to retrieve pellets.

There are two key findings, summarized nicely in the abstract. First, a tone-shock pairing that produced robust tone-elicited freezing in a small conditioning chamber completely failed to inhibit pellet retrieval in the naturalistic foraging task. The authors argue that this indicates that laboratory fear conditioning conditions elicit a type of learning not seen in natural settings. Second, animals that receive owl-shock pairing on the first day show robust fear to a tone presented on the testing day, despite never having heard the tone previously. The authors argue that this suggests that non-associative processes may be important for guiding fear behavior in the real-world. I enjoyed reading this manuscript, and I liked many aspects of the experiments. I appreciated the inclusion and reporting of data by sex, even though no sex differences were present for most of the results. More investigators should report data in this manner. I also appreciated that the authors measured things like path length, and latency to leave the nest, in addition to the time to retrieve pellets.

However, I do have some suggestions for strengthening the claims of the manuscript, especially given that they contradict the typical "party line" in the fear conditioning world (and just to be clear: I am completely fine with the idea that laboratory fear conditioning doesn't assess processing that is relevant to real-world conditions: this could explain one reason why it has been so hard to find new treatments for clinical disorders involving fear). These are described below.

The observation that tone-shock conditioning does not appear to alter behavior at all during testing is very interesting. Did the animals fail to make a tone-shock association when the tone and shock were administered in the foraging apparatus, or is an association acquired but simply not expressed when the competing demands of hunger and foraging are present during testing? This is an important question that the authors do not consider. I'd like to see a control where the authors try to tease this out. For example, the authors argue that small chambers limit the behavioral repertoire to freezing/not freezing. The authors could run a control in which animals are trained on the foraging task and administered tone-shock conditioning. Then, on the testing day, instead of turning on the tone and opening the gate 5s later, the tone could be presented and the gate could remain closed. The authors would then measure freezing in the smaller nest area. If freezing was observed, it would suggest that a tone-shock association was acquired, but that it is not expressed when foraging is available as a behavioral option. If freezing was not observed, this might suggest that no associative learning was acquired.

I'd also like the authors to explicitly state whether the animals in Figure 4 and Supplementary Figure 4 (where fear conditioning in the standard boxes was assessed) were food restricted in the same manner and for the same length of time as the animals in the foraging experiments. If they were not food restricted, then this control experiment needs to be repeated on food restricted animals. In my own lab, we find that food restricted animals generally freeze less in conditioning experiments than

non-restricted animals. The critical control is whether the tone and neck shock pairing produces freezing in food-restricted rats; it isn't completely clear that this is the control being reported. It is a possibility that food restricted animals minimize the use of freezing in their behavioral repertoire.

A more minor concern is that I wonder about the decibel level of the tone in the much larger foraging apparatus and whether the tone intensity at the time the animal receives the shock is lower than at the start of the apparatus, where the speakers are located, which might result in weaker associative fear learning than in a small conditioning chamber. (In contrast, the tone is uniformly loud in a small conditioning box). I suggest the authors measure the dB level of the tone in the trigger zone and compare this to the dB level in the nest area in the Methods.

A second minor point: I suggest that the authors move Figure 4 and its associated text to either the beginning of the results, or situated between what is currently the first and second section of results. As I read the results of the foraging experiments, I repeatedly thought that the fear conditioning with neck shock might be ineffective at producing learned fear—by presenting this first, I think readers will be more impressed with the absence of tone-shock conditioning in the foraging apparatus.

Regarding the second observation, that either owl-shock or tone-owl-shock pairings resulted in fear in the foraging task, I also find this very interesting. I think there is an additional control that could help with the interpretation of the results. It is clear that an owl without shock or shock without owl is sufficient to induce a fear-related reduction of pellet retrieval, even though both, by themselves, elicit robust unconditional fear responses (immediate fleeing to the nest area). It is also clear that is the owl and the shock, together, that supports fear learning/expression in the foraging environment. It seems critical, then, to understand the role of the owl. There are properties of the owl (distant threat, as well as predatory nature) that are not present in the neck shock (proximal threat, non-predatory). I'd like to know if the unique properties of the owl are necessary for sensitization. I'd like to see a control in which rats receive shock-shock conditioning (the first shock being triggered in lieu of the owl appearance). If the "predatory" aspect of threat or the combination of a proximal and distal threat are critical for the fear behavior during testing, then a shock-shock control should look like a tone-shock control and not exhibit fear to a novel tone. If instead it is simply sensitization by two USs occurring closely in time, then a shock-shock control should show fear to the novel tone (as the owl-shock group). I think this is important to know because it helps us think about whether standard laboratory conditioning can be adapted to better model real-world processes like sensitization (would two shocks given after a single CS be sufficient to do this?) or whether we need to move to completely new models (since owl cues are not as easily incorporated).

Reviewer #2 (Remarks to the Author):

This is a thought-provoking study investigating fear conditioning in a more naturalistic setting where rats must leave a safe nest area to forage for food in a larger arena. 1-trial conditioning is used throughout, to better simulate wild encounters with an attacking predator. Four groups are tested. A standard Pavlovian group pairs a simple tone with a mild shock to the back (T-S). Another group pairs the tone with a moving artificial owl stimulus, simulating an attack without contact (T-O). A third group pairs the tone with an owl+shock stimulus, simulating an attack with contact (T/O-S). A final group receives just the owl+shock US (O-S), simulating an attack with contact but no warning. Escape speeds and trajectories are measured during training, but the primary outcome is time to procure a food pellet during a 60s tone the next day. They find evidence for context conditioning with the owl+shock US, but not the owl or shock alone (latency to leave the nest). Further, when challenged with the tone, both groups trained with the owl+shock US initially retreat and take longer to procure a pellet, but this is not seen for groups trained with the owl or shock alone. A final experiment shows normal Pavlovian freezing using similar parameters in a T-S group trained and tested in standard,

small conditioning chambers. This work also indicates that pain inflicted by an outside agent is perceived as much more aversive than pain or the agent alone. The authors conclude that fear conditioning does not readily occur in naturalistic settings with many response choices, and instead, non-associative sensitization plays a more prominent role in defense.

This is a provocative study that challenges the exclusive use of simple (standard) procedures to understand fear learning and behavior in the real world. Work like this makes an important and needed contribution to current debates in the field about improving translational neuroscience for human anxiety. I do have some questions, however, about the main conclusions and suggest the authors address them to improve the impact of the paper.

1. The authors appear to favor an account where 1-trial associative fear conditioning doesn't occur in naturalistic settings, though they admit that it may just be very weak (lines 242-248). However, they lack the data to strongly support this conclusion, and some of the presented data may suggest the opposite. First, figures 3b&d(right) pretty clearly show a ceiling effect for rats in the T/O-S group that may prevent seeing an additional influence of Pavlovian memory on latency to procure the pellet (compared to O-S group). This ceiling effect also affects the correlations used to evaluate responding after different training ISIs. Second, escape speeds and trajectories are not reported for the tone test, which might show an additional effect of tone conditioning. Third, T-S rats trained in the foraging situation are never tested in the smaller fear conditioning chambers using the same dependent measure (freezing). It is possible that equivalent associative learning occurs in both situations but is expressed differently in the foraging arena. Last, 1-trial context conditioning was evident in two of the groups which shows that Pavlovian fear can be learned and expressed in the foraging situation provided that the US is strong enough (O-S compound).

2. The authors conclude that non-associative sensitization mediates fear-like responding to the tone after owl-shock encounters – mainly because the O-S group never experienced the tone until the test. This may be true, however, they also report significant context conditioning induced by the same treatment. Thus, it remains possible that the flight and delay in pellet procurement induced by the tone is due to associative context conditioning for O-S rats – similar to fear potentiated startle. Sensitization would be clearer if the same tone reaction for O-S rats transferred to a different context, but this was not tested.

3. Line 243: was freezing measured in the foraging arena during tone tests? If so, it would be good to report this too.

-Chris Cain

Responses to Reviewer Comments

We would like to express our sincere thanks to the reviewers for identifying areas of our manuscript that needed explanation and improvements. Below we detail our responses (in BLUE fonts) to all comments and suggestions and how they are reflected in the revision. We have also edited grammatical/typo errors and wording/readability wherever appropriate (e.g., conditioned response→conditional response). The revised manuscript shows all the changes (also in BLUE fonts) that have been made from the original version.

Reviewer #1

1) The observation that tone-shock conditioning does not appear to alter behavior at all during testing is very interesting. Did the animals fail to make a tone-shock association when the tone and shock were administered in the foraging apparatus, or is an association acquired but simply not expressed when the competing demands of hunger and foraging are present during testing? This is an important question that the authors do not consider. I'd like to see a control where the authors try to tease this out. For example, the authors argue that small chambers limit the behavioral repertoire to freezing/not freezing. The authors could run a control in which animals are trained on the foraging task and administered tone-shock conditioning. Then, on the testing day, instead of turning on the tone and opening the gate 5s later, the tone could be presented and the gate could remain closed. The authors would then measure freezing in the smaller nest area. If freezing was observed, it would suggest that a tone-shock association was acquired, but that is it not expressed when foraging is available as a behavioral option. If freezing was not observed, this might suggest that no associative learning was acquired.

Response: We thank the reviewer for suggesting this important control group in our study. Eight experimentally naive rats (4 females, 4 males) underwent the identical surgical, food restriction, habituation, baseline foraging and tone-shock conditioning procedures. After three pre-tone baseline trials (same as the previous T-S group), the tone CS was activated while the animals were inside the nest with the gateway closed. As can be seen from Author response image 1 below, during the 60 s of continuous tone, none of the animals exhibited reliable freezing behavior. When the gateway opened while the tone CS remained on, all rats readily entered the foraging arena and procured the pellet. Hence, fear behavior to the tone CS did not appear in the entire foraging apparatus, as assessed by the freezing response in the nest enclosure, the latency to forage when the gateway opened, and fleeing in the open arena. These animals were then given 2 additional tone-shock pairings (the same T-S procedure). Even after a total of 3 tone-shock pairings, the tone CS failed to reliably elicit freezing behavior inside the nest with the gateway closed and inhibit/delay the latency to forage when the gateway opened. These results further suggest that no associative learning to the tone CS was acquired in our naturalistic environment. The revised manuscript presents these new data in 'Tone testing inside the nest' section (pg. 8 and new Extended Data Fig. 6).

Author response image 1. Tone testing inside the nest and latency. **Top,** Mean (\pm SEM) and individual freezing data during the 1 min pre-tone and 1 min tone periods in the nest with the gateway closed. After a single tone-shock pairing (left), there was no significant increase in freezing from the pre-tone period to the tone period (Related-samples Wilcoxon signed rank test; $z = 0.524$, $p = 0.6$). After a total of three tone-shock pairings (right), there was still no reliable increase in freezing from the pre-tone period to the tone period (Related-samples Wilcoxon signed rank test; $z = -0.140$, $p = 0.889$). Insets show representative track plots of an animal inside the nest during the pre-tone and tone periods after one and three tone-shock pairings. **Bottom,** Mean (\pm SEM) and individual latencies to procure the food pellet during the baseline and tone foraging trials. After a single tone-shock pairing (left), there was no significant difference in the procurement latency before and during the tone (Related-samples Wilcoxon signed rank test; $z = -0.560$, $p = 0.575$). After a total of three tone-shock pairings (right), again there was no difference in the procurement latency before and during the tone (Related-samples Wilcoxon signed rank test; $z = -1.680$, $p = 0.093$). Insets show the three baseline trials.

2) I'd also like the authors to explicitly state whether the animals in Figure 4 and Supplementary Figure 4 (where fear conditioning in the standard boxes was assessed) were food restricted in the same manner and for the same length of time as the animals in the foraging experiments. If they were not food restricted, then this control experiment needs to be repeated on food restricted animals. In my own lab, we find that food restricted animals generally freeze less in conditioning experiments than non-restricted animals. The critical control is whether the tone and neck shock pairing produces freezing in food-restricted rats; it isn't completely clear that this is the control being reported. It is a possibility that food restricted animals minimize the use of freezing in their behavioral repertoire.

Response: The original Figure 4 animals were not food restricted based on the literature; for example, Maren & Fanselow (1998) used a similar one-trial learning procedure (a single 0.5 mA footshock) in food deprived (80-85% ad lib body weight) rats and showed that food deprived rats did not differ from non-deprived controls in freezing to the training context. Similarly, Brownlow et al. (2014) found that food deprived mice (65% ad lib body weight) did not show statistically different freezing levels to a tone CS (0.5 mA footshock; 2 pairings) compared to non-deprived mice. However, we agree with the reviewer that hunger motivation could be a confounding variable with the dorsal neck/body shock US. To remedy hunger and other procedural differences from obscuring the interpretation of previous 'Fear conditioning in a standard chamber' data, 8 other experimentally naïve rats (4 females and 4 males) underwent the same subcutaneous wire implant surgery, food restriction, habituation, and baseline foraging procedures, except they were presented with a tone CS-dorsal neck/body shock US pairing in a standard conditioning chamber. The side-by-side comparison of new (Author response image 2) and previous data (in RED box) show comparable postshock freezing and tone CS-elicited freezing levels. The revised manuscript now includes a new Fig. 4 and amended procedures in the Materials and Methods (pgs. 13-Subjects and 15-Behavioral Procedure for Standard Fear Conditioning).

Author response image 2. Auditory Fear conditioning in a standard chamber. **a**, Illustrations of a rat implanted with wires subcutaneously in the dorsal neck/body region undergoing successive days of habituation (10 min tethered, conditioning chamber), training (a single tone CS-shock US pairing), and tone testing (context shift). **b**, Mean (crimson line) and individual (gray lines) percent freezing data from 8 rats (4 females, 4 males) during training in context A: 3 min baseline (BL1, BL2, BL3); 23.1 s epoch of tone (T); 1 min postshock (PS). **c**, Mean and individual percent freezing data during tone testing in context B: 1 min baseline (BL1); 3 min tone (T1, T2, T3); 1 min post-tone (PT). **d**, Mean + SEM (bar) and individual (dots) percent freezing to tone CS before (Train, T) and after (Test, T1) undergoing auditory fear conditioning (paired t test; $t(7) = -3.188$, $p = 0.015$). *** $p < 0.001$.

3) I suggest the authors measure the dB level of the tone in the trigger zone and compare this to the dB level in the nest area in the Methods.

Response: The tone dB level in the nest (81 dB) is now mentioned in the revised manuscript (pg. 14; lines 361): "...3 kHz tone CS (80 dB measured from the trigger location; 81 dB measured within the nest area)..." Given that 'Tone testing inside the nest' (Comment 1 response) showed virtually no tone CS-evoked fear behavior, the 1 dB difference in the nest vs. out in the foraging zone cannot account for the lack of tone fear conditioning in a naturalistic environment.

4) A second minor point: I suggest that the authors move Figure 4 and its associated text to either the beginning of the results, or situated between what is currently the first and second section of results.

Response: We respectfully prefer to keep the original flow of our data presentation to highlight the study's central finding that traditional tone-shock paired animals (T-S group) do not readily show conditioned fear. As an alternative to the reviewer's suggestion, we have now referenced Figure 4 experiment in the introduction of the revised manuscript (pg. 4; lines 79-81) to give the readers a better understanding of the results: "Because the dorsal neck/body shock US has never been used before in fear research, its efficacy to support a single trial tone fear conditioning was also examined in a standard conditioning chamber."

5) I'd like to see a control in which rats receive shock-shock conditioning (the first shock being triggered in lieu of the owl appearance). If the "predatory" aspect of threat or the combination of a proximal and distal threat are critical for the fear behavior during testing, then a shock-shock control should look like a tone-shock control and not exhibit fear to a novel tone. If instead it is simply sensitization by two USs occurring closely in time, then a shock-shock control should show fear to the novel tone (as the owl-shock group). I think this is important to know because it helps us think about whether standard laboratory conditioning can be adapted to better model real-world processes like sensitization (would two shocks given after a single CS be sufficient to do this?) or whether we need to move to completely new models (since owl cues are not as easily incorporated).

Response: The reviewer suggests that shock-shock conditioning may have relevance towards interpreting the tone-shock (a single US) vs. owl-shock (a serial owl US and shock US) results of our study. The shock-shock conditioning was first employed in a human study by delivering a series of electric shock to the right foot and electric shock to the left foot to study conditioned withdrawal responses (Chmikhov, 1913). Perhaps, a modified shock-shock procedure might be possible to implement in standard fear conditioning models by inserting a clear-cut trace interval to separate the first (antecedent) footshock and the second (succeeding) footshock. However, in our owl-shock (as well as tone/owl-shock) condition, the 1-s shock was triggered 100-ms after the owl was activated while the rat was turning toward the nest (Fig. 2F). To match with the owl-shock's 100-ms ISI, the suggested shock-shock condition in our tethered shock system would be 100-ms shock followed by 1-s shock (a continuous 1.1-s shock), which is not significantly different from the current 1-s duration shock. Furthermore, our current headstage mount-tethered shock system cannot deliver two different intensities of shock with 100-ms ISI to the same dorsal neck/body region or two same intensity shocks with 100-ms ISI to two different body regions (necessary for the animals to distinguish two shock USs). We believe that the new results shown in the Author response image 1 above, however, may partly address the one US (tone-shock) vs. two US (owl-shock) question insightfully raised by the reviewer. Specifically, even after a total of three tone-shock pairings (three US episodes), tone CS-induced fear did not emerge. As we originally stated in the Discussion, the same 2.5 mA/1-s dorsal neck/body shock US may be interpreted as a life-or-death situation when naturally fleeing from an external threat agent (present study) versus a mere nociceptive situation in the absence of an external threat agent (standard fear conditioning studies).

In terms of how "standard laboratory conditioning can be adapted to better model" real-world danger scenarios, we suggest (pg. 11) an owl-shock like procedure can perhaps be introduced in standard conditioning chambers outfitted with overhead monitors to produce two-dimensional (2D) looming stimuli (e.g., rapidly expanding black disc) that can evoke freezing if the animal is distant from an enclosed shelter or fleeing if the animal is nearby an enclosed shelter (Yilmaz & Meister, 2013). In doing so, footshock can be delivered as the animals are either freezing or fleeing to 2D looming disc to potentially investigate, for example, nonassociative aspects of fear mechanisms and whether the same predatory strike evokes differential defensive responses on

animals engaged in the central amygdala-mediated freezing vs. the basolateral amygdala-mediated active avoidance (Cain & LeDoux, 2008; Choi, Cain & LeDoux, 2014). Incorporating dynamic external agents of danger into standard fear conditioning paradigms may lead to more realistic translational findings. However, it must also be considered that fear behaviors observed in an ecologically-relevant environment, where instinctive activities are unconstrained, might not be similarly observed in a standard operant chamber, where conditioning plays a disproportionately dominant role over instinctive fears, and vice versa (Thorndike, 1900; see pg. 3).

Reviewer #2

1) First, figures 3b&d(right) pretty clearly show a ceiling effect for rats in the T/O-S group that may prevent seeing an additional influence of Pavlovian memory on latency to procure the pellet (compared to O-S group). This ceiling effect also affects the correlations used to evaluate responding after different training ISIs.

Response: To show further comparisons of the O-S and T/O-S groups, additional tone testing data from the 3rd and 4th testing days have been added below (Author response image 3) and are now included in the revised manuscript (Extended Data Fig. 4). Note that animals underwent 3 tone tests daily until they successfully attained the pellet (mentioned in the original manuscript). There were no significant differences between the O-S and T/O-S groups on any of these measures. Additionally, the latencies to pellet procurement during the 3rd and 4th tone tests were not correlated with the tone-shock intervals during training even though there was more variability within the T/O-S groups' procurement times. These results suggest there seems not to be an additional influence of Pavlovian memory on the latency to procure the pellet in the T/O-S group.

Author response image 3. Extended tone testing information. Analyses of the third testing day (D-3) revealed there were no significant differences between the O-S and T/O-S groups in **a**, the mean latency (\pm SEM) to procure the pellet during the pre-tone baseline trials (Mann-Whitney U, $z = 1.304, p = 0.209$); **b**, the mean success rate and the latency to procure the pellet during the tone trial (Success Rate: Mann-Whitney U, $z = 0.63, p = 0.528$; Tone-Pellet Latency: Mann-Whitney U, $z = -0.638, p = 0.523$); and **c**, the correlation (Spearman's correlation coefficient) between the latency to procure the pellet during the tone and the time interval between the tone CS onset and shock onset (ISIs) during training. Same analyses of the fourth testing day (D-4) showed there were no significant group differences in **d**, the mean latency (\pm SEM) to procure the pellet during the pre-tone baseline trials (Mann-Whitney U, $z = 0, p = 1.0$); **e**, the mean success rate and the latency to procure the pellet during the tone trial (Success Rate: Mann-Whitney U, $z = 0.425, p = 0.671$; Tone-Pellet Latency: Mann-Whitney U, $z = -0.646, p = 0.518$); and **f**, the correlation between the latency to procure the pellet during the tone and the time interval between the tone CS onset and shock onset (ISIs) during training.

2) *Second, escape speeds and trajectories are not reported for the tone test, which might show an additional effect of tone conditioning.*

Response: The instantaneous speeds surrounding the tone presentation (Author response image 4) has now been added an Extended Data Fig. 5 (pg. 7). It represents the speeds from the track-plots presented in Fig. 3E. No significant differences were found between the O-S and T/O-S groups within -2 to 2 seconds around the tone presentation (tone activation was at 0). The variance of angles of the return trajectories during tone testing was also compared between the O-S and T/O-S groups with no significant differences found, indicating no evidence of additional effects of tone conditioning on the T/O-S group.

Author response image 4. Instantaneous speeds during tone testing and angle trajectories of escape. **a**, Mean (\pm SEM) speed of each rat as it approached and triggered the tone during the tone testing day. **b**, No significant difference was found between the speeds of the O-S and T/O-S groups (Mann-Whitney U, $z = -1.539$, $p = 0.124$). **c**, No significant differences were found between the O-S and T/O-S groups' return trajectories, i.e., the variance of angles (radians), to the nest during the tone testing (Mann-Whitney U, $z = -1.871$, $p = 0.064$).

3) *Third, T-S rats trained in the foraging situation are never tested in the smaller fear conditioning chambers using the same dependent measure (freezing). It is possible that equivalent associative learning occurs in both situations but is expressed differently in the foraging arena.*

Response: A similar comment was made by Reviewer 1 (Comment 1) and we now include the results of 'Tone testing inside the nest' experiment (Author response image 1) in the revised manuscript (pg. 8; new Extended Data Fig. 6).

4) *Last, 1-trial context conditioning was evident in two of the groups which shows that Pavlovian fear can be learned and expressed in the foraging situation provided that the US is strong enough (O-S compound).*

Response: In the original manuscript, we reported that owl-shock and tone/owl-shock animals took longer to procure the pellet on the first pre-tone trial of the test day than the other groups, indicative of contextual conditioning. Contrary to standard fear conditioning studies, however, we also found the tone-shock animals produce neither auditory nor background contextual fear conditioning despite the 2.5 mA shock US evoking robust fleeing UR. The author response image 1 above (new Extended Data Fig. 6) now shows absence of fear conditioning even after a total of 3 pairings of tone CS with 2.5 mA shock US. The fact that context conditioning occurred after a single nociceptive experience preceded by an owl, but not after three nociceptive experiences preceded by a tone CS, further suggests that the same 2.5 mA/1-s dorsal neck/body shock US may be interpreted as a life-or-death situation when naturally fleeing from an external threat agent (present study) versus a mere nociceptive situation in the absence of an external threat agent (standard fear conditioning studies). As previously

mentioned in the Discussion, there is a possibility of rapid associative processes at work (Fanselow, 2018) between the owl and shock stimuli, resulting in a more ethological US that can perhaps better support contextual fear learning, since neither the tone-shock group nor the tone-owl group showed evidence of context conditioning. However, “a rapid owl-shock association nevertheless cannot explain the O-S animals’ subsequent fleeing behavior to a novel tone (in the absence of owl), which likely reflects nonassociative fear.” Future studies will need to delineate the nature and mechanisms of nonassociative fear responses in the presence of predators in animals and perpetrators in humans that normally inflict bodily injuries in nature. As responded to the Reviewer 1’s comment 5 above, we now mention the following in the Discussion (pg. 11): The present owl-shock like procedure can perhaps be introduced in standard conditioning chambers outfitted with overhead monitors to produce two-dimensional (2D) looming stimuli (e.g., rapidly expanding black disc) that can evoke freezing if the animal is distant from an enclosed shelter or fleeing if the animal is nearby an enclosed shelter (Yilmaz & Meister, 2013). In doing so, footshock can be delivered as the animals are either freezing or fleeing to 2D looming disc to potentially investigate, for example, nonassociative aspects of fear mechanisms and whether the same predatory strike evokes differential defensive responses on animals engaged in the central amygdala-mediated freezing vs. the basolateral amygdala-mediated active avoidance (Cain & LeDoux, 2008; Choi, Cain & LeDoux, 2014). Incorporating dynamic external agents of danger into standard fear conditioning paradigms may lead to more realistic translational findings. However, it must also be considered that fear behaviors observed in an ecologically-relevant environment, where instinctive activities are unconstrained, might not be similarly observed in a standard operant chamber, where conditioning plays a disproportionately dominant role over instinctive fears, and vice versa (Thorndike, 1900; see pg. 3).

5) *...it remains possible that the flight and delay in pellet procurement induced by the tone is due to associative context conditioning for O-S rats – similar to fear potentiated startle. Sensitization would be clearer if the same tone reaction for O-S rats transferred to a different context, but this was not tested.*

Response: We now mention the possibility that the novel tone-evoked flight behavior in O-S rats might represent a nonassociative process driven by associative context conditioning in the Discussion (pg. 9-10, lines 230-234). Specifically, we suggest that the O-S data, i.e., a novel tone eliciting similar fleeing behavior caused by the owl-shock experience the previous day, better represent pseudo-conditioning than sensitization (both nonassociative processes). Procedurally, pseudo-conditioning refers to a UR-like behavior emerging to a novel stimulus after mere exposure to a biologically significant US, whereas sensitization refers to amplified response with repeated presentation of the same stimulus (e.g., Bouton, 2007; Mackintosh, 1974). The contextual fear observed in owl-shock (as well as tone/owl-shock) animals is also consistent with the finding that pseudo-conditioning transpires from conditioning of the context (Sheafor, 1975). We thus replaced all previous mentions of sensitization with a possible nonassociative pseudo-conditioning process throughout the revised manuscript. Regardless of the precise mechanisms responsible for the O-S data, we wish to emphasize that the central finding of the study is that the traditional tone-shock pairing animals do not readily show conditioned fear in an ecologically-relevant scenario.

6) Line 243: was freezing measured in the foraging arena during tone tests? If so, it would be good to report this too.

Response: We agree and now have added the following to the revised manuscript (pg. 7; line 149-151): “No freezing (as measured by the ANY-maze tracking software with a 2 s threshold) was detected in the foraging arena during the tone presentations.”

Response Letter References

- Bouton, M. E. (2007). *Learning and Behavior*. Sinauer Associates
- Brownlow, M. L., Joly-Amado, A., Azam, S., Elza, M., Selenica, M. L., Pappas, C., . . . Morgan, D. (2014). Partial rescue of memory deficits induced by calorie restriction in a mouse model of tau deposition. *Behav Brain Res*, 271, 79-88.
- Cain, C. K., & LeDoux, J. E. (2008). Brain mechanisms of Pavlovian and instrumental aversive conditioning. *Handbook Behav Neurosci*, 17, 103-124.
- Choi, J. S., Cain, C. K., & LeDoux, J. E. (2014). The role of amygdala nuclei in the expression of auditory signaled two-way active avoidance in rats. *Learn Mem*, 17, 139-147.
- Chmikhov, A. M. (1913). *The interrelations between two qualitatively associated reflexes of the symmetrical extremities in human beings*. St. Petersburg.
- Fanselow, M. S. (2018). The Role of Learning in Threat Imminence and Defensive Behaviors. *Curr Opin Behav Sci*, 24, 44-49.
- Mackintosh, N. J. (1974). *The psychology of animal learning*. Academic Press.
- Maren, S., & Fanselow, M. S. (1998). Appetitive motivational states differ in their ability to augment aversive fear conditioning in rats (*Rattus norvegicus*). *J Exp Psychol Anim Behav Process*, 24(3), 369-373.
- Mattson, B. J., Koya, E., Simmons, D. E., Mitchell, T. B., Berkow, A., Crombag, H. S., & Hope, B. T. (2008). Context-specific sensitization of cocaine-induced locomotor activity and associated neuronal ensembles in rat nucleus accumbens. *Eur J Neurosci*, 27(1), 202-212.
- Sheafor, P. J. (1975). Pseudoconditioned jaw movements of the rabbit reflect associations conditioned to contextual background cues. *J Exp Psychol: Anim Behav Process*, 1(3), 245.
- Stewart, J., & Badiani, A. (1993). Tolerance and sensitization to the behavioral effects of drugs. *Behav Pharmacol*, 4(4), 289-312.
- Thorndike, E. (1900). Biological Lectures from the Marine Laboratory at Woods' Holl, USA, for 1899. *Nature*, 62(1609).
- Yilmaz, M., & Meister, M. (2013). Rapid innate defensive responses of mice to looming visual stimuli. *Curr Biol*, 23(20), 2011-2015.

REVIEWERS' COMMENTS:

Reviewer #1 (Remarks to the Author):

I carefully read the revised manuscript and the responses to both sets of reviewer comments, and I find that the authors have thoroughly and effectively addressed both my concerns and those of the second reviewer. I believe the revised manuscript more clearly shows that an owl-shock pairing, but not tone-shock pairing, inhibits foraging behavior. The manuscript also highlights the important finding that the ability of a single tone-shock pairing to generate associative fear memory is context-dependent: conditional freezing is observed in standard fear conditioning boxes in food-deprived rats, but no evidence of fear is present when the same pairing is administered in a naturalistic environment.

I encourage the authors to publish their findings that the neck shock produces conditional freezing when the tone-shock pairing was administered in rats given ad lib chow (not food-deprived). This was removed from the manuscript, but it is an important validation of the technique, even though it is not directly relevant to the findings here. I hope they choose to include these data in a subsequent paper.

Reviewer #2 (Remarks to the Author):

The authors were very responsive to previous reviews and the manuscript is substantially improved. Additional experiments and analyses were conducted that address my previous concerns and strongly support the authors' conclusions. This is a very interesting and novel study that may change how researchers design and interpret fear conditioning studies. I recommend that the manuscript be accepted for publication in its current form.

Responses to Reviewer Comments

We would like to express our sincere thanks to the reviewers for identifying areas of our manuscript that needed explanation and improvements. Below we detail our responses (in BLUE fonts) to all comments and suggestions and how they are reflected in the revision. We have also edited grammatical/typo errors and wording/readability wherever appropriate (e.g., conditioned response→conditional response). The revised manuscript shows all the changes (also in BLUE fonts) that have been made from the original version.

Reviewer #1

1) The observation that tone-shock conditioning does not appear to alter behavior at all during testing is very interesting. Did the animals fail to make a tone-shock association when the tone and shock were administered in the foraging apparatus, or is an association acquired but simply not expressed when the competing demands of hunger and foraging are present during testing? This is an important question that the authors do not consider. I'd like to see a control where the authors try to tease this out. For example, the authors argue that small chambers limit the behavioral repertoire to freezing/not freezing. The authors could run a control in which animals are trained on the foraging task and administered tone-shock conditioning. Then, on the testing day, instead of turning on the tone and opening the gate 5s later, the tone could be presented and the gate could remain closed. The authors would then measure freezing in the smaller nest area. If freezing was observed, it would suggest that a tone-shock association was acquired, but that is it not expressed when foraging is available as a behavioral option. If freezing was not observed, this might suggest that no associative learning was acquired.

Response: We thank the reviewer for suggesting this important control group in our study. Eight experimentally naive rats (4 females, 4 males) underwent the identical surgical, food restriction, habituation, baseline foraging and tone-shock conditioning procedures. After three pre-tone baseline trials (same as the previous T-S group), the tone CS was activated while the animals were inside the nest with the gateway closed. As can be seen from Author response image 1 below, during the 60 s of continuous tone, none of the animals exhibited reliable freezing behavior. When the gateway opened while the tone CS remained on, all rats readily entered the foraging arena and procured the pellet. Hence, fear behavior to the tone CS did not appear in the entire foraging apparatus, as assessed by the freezing response in the nest enclosure, the latency to forage when the gateway opened, and fleeing in the open arena. These animals were then given 2 additional tone-shock pairings (the same T-S procedure). Even after a total of 3 tone-shock pairings, the tone CS failed to reliably elicit freezing behavior inside the nest with the gateway closed and inhibit/delay the latency to forage when the gateway opened. These results further suggest that no associative learning to the tone CS was acquired in our naturalistic environment. The revised manuscript presents these new data in 'Tone testing inside the nest' section (pg. 8 and new Extended Data Fig. 6).

Author response image 1. Tone testing inside the nest and latency. **Top,** Mean (\pm SEM) and individual freezing data during the 1 min pre-tone and 1 min tone periods in the nest with the gateway closed. After a single tone-shock pairing (left), there was no significant increase in freezing from the pre-tone period to the tone period (Related-samples Wilcoxon signed rank test; $z = 0.524$, $p = 0.6$). After a total of three tone-shock pairings (right), there was still no reliable increase in freezing from the pre-tone period to the tone period (Related-samples Wilcoxon signed rank test; $z = -0.140$, $p = 0.889$). Insets show representative track plots of an animal inside the nest during the pre-tone and tone periods after one and three tone-shock pairings. **Bottom,** Mean (\pm SEM) and individual latencies to procure the food pellet during the baseline and tone foraging trials. After a single tone-shock pairing (left), there was no significant difference in the procurement latency before and during the tone (Related-samples Wilcoxon signed rank test; $z = -0.560$, $p = 0.575$). After a total of three tone-shock pairings (right), again there was no difference in the procurement latency before and during the tone (Related-samples Wilcoxon signed rank test; $z = -1.680$, $p = 0.093$). Insets show the three baseline trials.

2) I'd also like the authors to explicitly state whether the animals in Figure 4 and Supplementary Figure 4 (where fear conditioning in the standard boxes was assessed) were food restricted in the same manner and for the same length of time as the animals in the foraging experiments. If they were not food restricted, then this control experiment needs to be repeated on food restricted animals. In my own lab, we find that food restricted animals generally freeze less in conditioning experiments than non-restricted animals. The critical control is whether the tone and neck shock pairing produces freezing in food-restricted rats; it isn't completely clear that this is the control being reported. It is a possibility that food restricted animals minimize the use of freezing in their behavioral repertoire.

Response: The original Figure 4 animals were not food restricted based on the literature; for example, Maren & Fanselow (1998) used a similar one-trial learning procedure (a single 0.5 mA footshock) in food deprived (80-85% ad lib body weight) rats and showed that food deprived rats did not differ from non-deprived controls in freezing to the training context. Similarly, Brownlow et al. (2014) found that food deprived mice (65% ad lib body weight) did not show statistically different freezing levels to a tone CS (0.5 mA footshock; 2 pairings) compared to non-deprived mice. However, we agree with the reviewer that hunger motivation could be a confounding variable with the dorsal neck/body shock US. To remedy hunger and other procedural differences from obscuring the interpretation of previous 'Fear conditioning in a standard chamber' data, 8 other experimentally naïve rats (4 females and 4 males) underwent the same subcutaneous wire implant surgery, food restriction, habituation, and baseline foraging procedures, except they were presented with a tone CS-dorsal neck/body shock US pairing in a standard conditioning chamber. The side-by-side comparison of new (Author response image 2) and previous data (in RED box) show comparable postshock freezing and tone CS-elicited freezing levels. The revised manuscript now includes a new Fig. 4 and amended procedures in the Materials and Methods (pgs. 13-Subjects and 15-Behavioral Procedure for Standard Fear Conditioning).

Author response image 2. Auditory Fear conditioning in a standard chamber. **a**, Illustrations of a rat implanted with wires subcutaneously in the dorsal neck/body region undergoing successive days of habituation (10 min tethered, conditioning chamber), training (a single tone CS-shock US pairing), and tone testing (context shift). **b**, Mean (crimson line) and individual (gray lines) percent freezing data from 8 rats (4 females, 4 males) during training in context A: 3 min baseline (BL1, BL2, BL3); 23.1 s epoch of tone (T); 1 min postshock (PS). **c**, Mean and individual percent freezing data during tone testing in context B: 1 min baseline (BL1); 3 min tone (T1, T2, T3); 1 min post-tone (PT). **d**, Mean + SEM (bar) and individual (dots) percent freezing to tone CS before (Train, T) and after (Test, T1) undergoing auditory fear conditioning (paired t test; $t(7) = -3.188$, $p = 0.015$). *** $p < 0.001$.

3) I suggest the authors measure the dB level of the tone in the trigger zone and compare this to the dB level in the nest area in the Methods.

Response: The tone dB level in the nest (81 dB) is now mentioned in the revised manuscript (pg. 14; lines 361): "...3 kHz tone CS (80 dB measured from the trigger location; 81 dB measured within the nest area)..." Given that 'Tone testing inside the nest' (Comment 1 response) showed virtually no tone CS-evoked fear behavior, the 1 dB difference in the nest vs. out in the foraging zone cannot account for the lack of tone fear conditioning in a naturalistic environment.

4) A second minor point: I suggest that the authors move Figure 4 and its associated text to either the beginning of the results, or situated between what is currently the first and second section of results.

Response: We respectfully prefer to keep the original flow of our data presentation to highlight the study's central finding that traditional tone-shock paired animals (T-S group) do not readily show conditioned fear. As an alternative to the reviewer's suggestion, we have now referenced Figure 4 experiment in the introduction of the revised manuscript (pg. 4; lines 79-81) to give the readers a better understanding of the results: "Because the dorsal neck/body shock US has never been used before in fear research, its efficacy to support a single trial tone fear conditioning was also examined in a standard conditioning chamber."

5) I'd like to see a control in which rats receive shock-shock conditioning (the first shock being triggered in lieu of the owl appearance). If the "predatory" aspect of threat or the combination of a proximal and distal threat are critical for the fear behavior during testing, then a shock-shock control should look like a tone-shock control and not exhibit fear to a novel tone. If instead it is simply sensitization by two USs occurring closely in time, then a shock-shock control should show fear to the novel tone (as the owl-shock group). I think this is important to know because it helps us think about whether standard laboratory conditioning can be adapted to better model real-world processes like sensitization (would two shocks given after a single CS be sufficient to do this?) or whether we need to move to completely new models (since owl cues are not as easily incorporated).

Response: The reviewer suggests that shock-shock conditioning may have relevance towards interpreting the tone-shock (a single US) vs. owl-shock (a serial owl US and shock US) results of our study. The shock-shock conditioning was first employed in a human study by delivering a series of electric shock to the right foot and electric shock to the left foot to study conditioned withdrawal responses (Chmikhov, 1913). Perhaps, a modified shock-shock procedure might be possible to implement in standard fear conditioning models by inserting a clear-cut trace interval to separate the first (antecedent) footshock and the second (succeeding) footshock. However, in our owl-shock (as well as tone/owl-shock) condition, the 1-s shock was triggered 100-ms after the owl was activated while the rat was turning toward the nest (Fig. 2F). To match with the owl-shock's 100-ms ISI, the suggested shock-shock condition in our tethered shock system would be 100-ms shock followed by 1-s shock (a continuous 1.1-s shock), which is not significantly different from the current 1-s duration shock. Furthermore, our current headstage mount-tethered shock system cannot deliver two different intensities of shock with 100-ms ISI to the same dorsal neck/body region or two same intensity shocks with 100-ms ISI to two different body regions (necessary for the animals to distinguish two shock USs). We believe that the new results shown in the Author response image 1 above, however, may partly address the one US (tone-shock) vs. two US (owl-shock) question insightfully raised by the reviewer. Specifically, even after a total of three tone-shock pairings (three US episodes), tone CS-induced fear did not emerge. As we originally stated in the Discussion, the same 2.5 mA/1-s dorsal neck/body shock US may be interpreted as a life-or-death situation when naturally fleeing from an external threat agent (present study) versus a mere nociceptive situation in the absence of an external threat agent (standard fear conditioning studies).

In terms of how "standard laboratory conditioning can be adapted to better model" real-world danger scenarios, we suggest (pg. 11) an owl-shock like procedure can perhaps be introduced in standard conditioning chambers outfitted with overhead monitors to produce two-dimensional (2D) looming stimuli (e.g., rapidly expanding black disc) that can evoke freezing if the animal is distant from an enclosed shelter or fleeing if the animal is nearby an enclosed shelter (Yilmaz & Meister, 2013). In doing so, footshock can be delivered as the animals are either freezing or fleeing to 2D looming disc to potentially investigate, for example, nonassociative aspects of fear mechanisms and whether the same predatory strike evokes differential defensive responses on

animals engaged in the central amygdala-mediated freezing vs. the basolateral amygdala-mediated active avoidance (Cain & LeDoux, 2008; Choi, Cain & LeDoux, 2014). Incorporating dynamic external agents of danger into standard fear conditioning paradigms may lead to more realistic translational findings. However, it must also be considered that fear behaviors observed in an ecologically-relevant environment, where instinctive activities are unconstrained, might not be similarly observed in a standard operant chamber, where conditioning plays a disproportionately dominant role over instinctive fears, and vice versa (Thorndike, 1900; see pg. 3).

Reviewer #2

1) First, figures 3b&d(right) pretty clearly show a ceiling effect for rats in the T/O-S group that may prevent seeing an additional influence of Pavlovian memory on latency to procure the pellet (compared to O-S group). This ceiling effect also affects the correlations used to evaluate responding after different training ISIs.

Response: To show further comparisons of the O-S and T/O-S groups, additional tone testing data from the 3rd and 4th testing days have been added below (Author response image 3) and are now included in the revised manuscript (Extended Data Fig. 4). Note that animals underwent 3 tone tests daily until they successfully attained the pellet (mentioned in the original manuscript). There were no significant differences between the O-S and T/O-S groups on any of these measures. Additionally, the latencies to pellet procurement during the 3rd and 4th tone tests were not correlated with the tone-shock intervals during training even though there was more variability within the T/O-S groups' procurement times. These results suggest there seems not to be an additional influence of Pavlovian memory on the latency to procure the pellet in the T/O-S group.

Author response image 3. Extended tone testing information. Analyses of the third testing day (D-3) revealed there were no significant differences between the O-S and T/O-S groups in **a**, the mean latency (\pm SEM) to procure the pellet during the pre-tone baseline trials (Mann-Whitney U, $z = 1.304$, $p = 0.209$); **b**, the mean success rate and the latency to procure the pellet during the tone trial (Success Rate: Mann-Whitney U, $z = 0.63$, $p = 0.528$; Tone-Pellet Latency: Mann-Whitney U, $z = -0.638$, $p = 0.523$); and **c**, the correlation (Spearman's correlation coefficient) between the latency to procure the pellet during the tone and the time interval between the tone CS onset and shock onset (ISIs) during training. Same analyses of the fourth testing day (D-4) showed there were no significant group differences in **d**, the mean latency (\pm SEM) to procure the pellet during the pre-tone baseline trials (Mann-Whitney U, $z = 0$, $p = 1.0$); **e**, the mean success rate and the latency to procure the pellet during the tone trial (Success Rate: Mann-Whitney U, $z = 0.425$, $p = 0.671$; Tone-Pellet Latency: Mann-Whitney U, $z = -0.646$, $p = 0.518$); and **f**, the correlation between the latency to procure the pellet during the tone and the time interval between the tone CS onset and shock onset (ISIs) during training.

2) Second, escape speeds and trajectories are not reported for the tone test, which might show an additional effect of tone conditioning.

Response: The instantaneous speeds surrounding the tone presentation (Author response image 4) has now been added an Extended Data Fig. 5 (pg. 7). It represents the speeds from the track-plots presented in Fig. 3E. No significant differences were found between the O-S and T/O-S groups within -2 to 2 seconds around the tone presentation (tone activation was at 0). The variance of angles of the return trajectories during tone testing was also compared between the O-S and T/O-S groups with no significant differences found, indicating no evidence of additional effects of tone conditioning on the T/O-S group.

Author response image 4. Instantaneous speeds during tone testing and angle trajectories of escape. **a**, Mean (\pm SEM) speed of each rat as it approached and triggered the tone during the tone testing day. **b**, No significant difference was found between the speeds of the O-S and T/O-S groups (Mann-Whitney U, $z = -1.539$, $p = 0.124$). **c**, No significant differences were found between the O-S and T/O-S groups' return trajectories, i.e., the variance of angles (radians), to the nest during the tone testing (Mann-Whitney U, $z = -1.871$, $p = 0.064$).

3) Third, T-S rats trained in the foraging situation are never tested in the smaller fear conditioning chambers using the same dependent measure (freezing). It is possible that equivalent associative learning occurs in both situations but is expressed differently in the foraging arena.

Response: A similar comment was made by Reviewer 1 (Comment 1) and we now include the results of 'Tone testing inside the nest' experiment (Author response image 1) in the revised manuscript (pg. 8; new Extended Data Fig. 6).

4) Last, 1-trial context conditioning was evident in two of the groups which shows that Pavlovian fear can be learned and expressed in the foraging situation provided that the US is strong enough (O-S compound).

Response: In the original manuscript, we reported that owl-shock and tone/owl-shock animals took longer to procure the pellet on the first pre-tone trial of the test day than the other groups, indicative of contextual conditioning. Contrary to standard fear conditioning studies, however, we also found the tone-shock animals produce neither auditory nor background contextual fear conditioning despite the 2.5 mA shock US evoking robust fleeing UR. The author response image 1 above (new Extended Data Fig. 6) now shows absence of fear conditioning even after a total of 3 pairings of tone CS with 2.5 mA shock US. The fact that context conditioning occurred after a single nociceptive experience preceded by an owl, but not after three nociceptive experiences preceded by a tone CS, further suggests that the same 2.5 mA/1-s dorsal neck/body shock US may be interpreted as a life-or-death situation when naturally fleeing from an external threat agent (present study) versus a mere nociceptive situation in the absence of an external threat agent (standard fear conditioning studies). As previously

mentioned in the Discussion, there is a possibility of rapid associative processes at work (Fanselow, 2018) between the owl and shock stimuli, resulting in a more ethological US that can perhaps better support contextual fear learning, since neither the tone-shock group nor the tone-owl group showed evidence of context conditioning. However, “a rapid owl-shock association nevertheless cannot explain the O-S animals’ subsequent fleeing behavior to a novel tone (in the absence of owl), which likely reflects nonassociative fear.” Future studies will need to delineate the nature and mechanisms of nonassociative fear responses in the presence of predators in animals and perpetrators in humans that normally inflict bodily injuries in nature. As responded to the Reviewer 1’s comment 5 above, we now mention the following in the Discussion (pg. 11): The present owl-shock like procedure can perhaps be introduced in standard conditioning chambers outfitted with overhead monitors to produce two-dimensional (2D) looming stimuli (e.g., rapidly expanding black disc) that can evoke freezing if the animal is distant from an enclosed shelter or fleeing if the animal is nearby an enclosed shelter (Yilmaz & Meister, 2013). In doing so, footshock can be delivered as the animals are either freezing or fleeing to 2D looming disc to potentially investigate, for example, nonassociative aspects of fear mechanisms and whether the same predatory strike evokes differential defensive responses on animals engaged in the central amygdala-mediated freezing vs. the basolateral amygdala-mediated active avoidance (Cain & LeDoux, 2008; Choi, Cain & LeDoux, 2014). Incorporating dynamic external agents of danger into standard fear conditioning paradigms may lead to more realistic translational findings. However, it must also be considered that fear behaviors observed in an ecologically-relevant environment, where instinctive activities are unconstrained, might not be similarly observed in a standard operant chamber, where conditioning plays a disproportionately dominant role over instinctive fears, and vice versa (Thorndike, 1900; see pg. 3).

5) *...it remains possible that the flight and delay in pellet procurement induced by the tone is due to associative context conditioning for O-S rats – similar to fear potentiated startle. Sensitization would be clearer if the same tone reaction for O-S rats transferred to a different context, but this was not tested.*

Response: We now mention the possibility that the novel tone-evoked flight behavior in O-S rats might represent a nonassociative process driven by associative context conditioning in the Discussion (pg. 9-10, lines 230-234). Specifically, we suggest that the O-S data, i.e., a novel tone eliciting similar fleeing behavior caused by the owl-shock experience the previous day, better represent pseudo-conditioning than sensitization (both nonassociative processes). Procedurally, pseudo-conditioning refers to a UR-like behavior emerging to a novel stimulus after mere exposure to a biologically significant US, whereas sensitization refers to amplified response with repeated presentation of the same stimulus (e.g., Bouton, 2007; Mackintosh, 1974). The contextual fear observed in owl-shock (as well as tone/owl-shock) animals is also consistent with the finding that pseudo-conditioning transpires from conditioning of the context (Sheafor, 1975). We thus replaced all previous mentions of sensitization with a possible nonassociative pseudo-conditioning process throughout the revised manuscript. Regardless of the precise mechanisms responsible for the O-S data, we wish to emphasize that the central finding of the study is that the traditional tone-shock pairing animals do not readily show conditioned fear in an ecologically-relevant scenario.

6) Line 243: was freezing measured in the foraging arena during tone tests? If so, it would be good to report this too.

Response: We agree and now have added the following to the revised manuscript (pg. 7; line 149-151): “No freezing (as measured by the ANY-maze tracking software with a 2 s threshold) was detected in the foraging arena during the tone presentations.”

Response Letter References

- Bouton, M. E. (2007). *Learning and Behavior*. Sinauer Associates
- Brownlow, M. L., Joly-Amado, A., Azam, S., Elza, M., Selenica, M. L., Pappas, C., . . . Morgan, D. (2014). Partial rescue of memory deficits induced by calorie restriction in a mouse model of tau deposition. *Behav Brain Res*, 271, 79-88.
- Cain, C. K., & LeDoux, J. E. (2008). Brain mechanisms of Pavlovian and instrumental aversive conditioning. *Handbook Behav Neurosci*, 17, 103-124.
- Choi, J. S., Cain, C. K., & LeDoux, J. E. (2014). The role of amygdala nuclei in the expression of auditory signaled two-way active avoidance in rats. *Learn Mem*, 17, 139-147.
- Chmikhov, A. M. (1913). *The interrelations between two qualitatively associated reflexes of the symmetrical extremities in human beings*. St. Petersburg.
- Fanselow, M. S. (2018). The Role of Learning in Threat Imminence and Defensive Behaviors. *Curr Opin Behav Sci*, 24, 44-49.
- Mackintosh, N. J. (1974). *The psychology of animal learning*. Academic Press.
- Maren, S., & Fanselow, M. S. (1998). Appetitive motivational states differ in their ability to augment aversive fear conditioning in rats (*Rattus norvegicus*). *J Exp Psychol Anim Behav Process*, 24(3), 369-373.
- Mattson, B. J., Koya, E., Simmons, D. E., Mitchell, T. B., Berkow, A., Crombag, H. S., & Hope, B. T. (2008). Context-specific sensitization of cocaine-induced locomotor activity and associated neuronal ensembles in rat nucleus accumbens. *Eur J Neurosci*, 27(1), 202-212.
- Sheafor, P. J. (1975). Pseudoconditioned jaw movements of the rabbit reflect associations conditioned to contextual background cues. *J Exp Psychol: Anim Behav Process*, 1(3), 245.
- Stewart, J., & Badiani, A. (1993). Tolerance and sensitization to the behavioral effects of drugs. *Behav Pharmacol*, 4(4), 289-312.
- Thorndike, E. (1900). Biological Lectures from the Marine Laboratory at Woods' Holl, USA, for 1899. *Nature*, 62(1609).
- Yilmaz, M., & Meister, M. (2013). Rapid innate defensive responses of mice to looming visual stimuli. *Curr Biol*, 23(20), 2011-2015.